# BOBA: Dynamic Bayesian Optimization through Bayesian Active Inference

## Abstract

Dynamic black-box optimization presents significant challenges for Bayesian Optimization (BO), as the objective function evolves over time, causing optimal locations to shift continuously. Existing dynamic BO (DBO) methods using standard acquisition functions such as Upper Confidence Bound (UCB) fail to explicitly account for temporal variations, leading to suboptimal sample allocation and poor tracking of moving optima. Here, we propose BOBA (**B**ayesian **O**ptimization through **B**ayesian **A**ctive Inference), a novel acquisition function inspired by free energy principles from active inference that explicitly minimizes predictive uncertainty about future states in dynamic environments. BOBA extends traditional acquisition functions by incorporating a forward-looking uncertainty quantification that estimates uncertainty in function changes, enabling more informed exploration-exploitation trade-offs in non-stationary settings. We evaluate BOBA on synthetic dynamic benchmarks, comparing against state-of-the-art DBO methods. Our experiments demonstrate that BOBA significantly improves regret in query-restricted settings, while remaining competitive in time-limited settings. We further analyze variants of BOBA with different exploration strategies, showing how the exploration-exploitation balance can be tuned for different types of dynamic functions. This work contributes both a free energy-based acquisition function for DBO and insights into how active inference principles can enhance optimization in non-stationary environments, with implications for real-time applications requiring continuous adaptation.

## 1 Introduction

In real-world applications, many problems require optimization to maximize outcomes (Afergan et al., 2014; Zhang et al., 2022; Cole et al., 2024). Machine learning has been able to enhance several fields based on optimizing black box functions $f(x)$ such as robot control (Wagener et al., 2019), user interfaces in automated vehicles (Jansen et al., 2025), and even deep brain stimulation (Cole et al., 2024). The most common methods used to create an optimal model consist of reinforcement learning (RL) and Bayesian optimization (BO) (Bian & Priyadarshi, 2024).

RL has proven effective at discovering an optimal policy for costly-to-evaluate black box functions by trying to maximize reward for the current $f(x)$ (Pires et al., 2022; Mahmud et al., 2018). Moreover, to discover an optimal policy, several iterations are required to explore $f(x)$ in terms of policies and the potential reward each policy can generate. Unfortunately, if $f(x)$ is costly to sample and is time-dependent, such as physiological data, convergence on the optimal policies becomes increasingly difficult as the optimal will change with time (Padakandla, 2022).

BO is better suited to optimize black box models when $f(x)$ is dependent on time due to its ability to model $f(x)$ in real-time. Moreover, BO is ideal for costly black-box function evaluations because it uses acquisition functions to guide sample selection efficiently, minimizing the number of evaluations needed compared to gradient-free methods like evolutionary algorithms or grid search. Alternatively, standard BO still has its limitations when $f(x, t)$ as time can only be observed in the present, reflected from the past and estimated in the future. For these functions, dynamic BO (DBO) has been used to identify the optimum point over time (Bogunovic et al., 2016; Bardou et al., 2024; Bardou & Thiran, 2025; Nyikosa et al., 2018). Nevertheless, one under-explored area of DBO is the

development of acquisition functions specifically designed for dynamic settings and their ability to infer temporal changes in $f(x, t)$. .

In this paper, we introduce BOBA, the first acquisition function that exploits free energy principles derived from active inference (AIF) to determine the next query for DBO, jointly optimizing intrinsic and extrinsic value. AIF describes optimal behavior performed by agents in a fixed environment, where the agent will try to minimize probability of uncertainty while maximizing probability of a desired outcome (Parr et al., 2020; Costa et al., 2020; Friston et al., 2023a). By unifying the two computation paradigms for the first time, we propose this acquisition function that can enhance DBO optimization by minimizing uncertainty while also giving avenues for active inference researchers to estimate a generative model in real time.

In this paper, the main contribution is the BOBA function, which is evaluated against and combined with current DBO models.

## 2 PRELIMINARIES

### 2.1 BAYESIAN OPTIMIZATION

In this paper, we focus on Bayesian optimization (BO), a framework that models objective functions $f(x)$ for the purpose of finding the minimum output across all inputs:

$$\min_{x \in \mathbf{S}} f(x) \tag{1}$$

Where $\mathbf{S}$ contains all possible inputs for $x$. The Gaussian Process (GP) is most widely utilized in BO frameworks as a probabilistic model that tries to replicate $f(x)$ with normally distributed uncertainties for $x_i \in \mathbf{S}$. A $GP(\mu(x), \mathbf{K}(x, x'))$ has a mean function $\mu(x) = 0$ that is assumed to be zero and a $n \times n$ covariance kernel $\mathbf{k}(\mathbf{X}, \mathbf{X}) = \mathbf{k}(x_i, x_j)_{i,j \leq n}$. Given $x_i \in \mathbf{S}$ and a dataset of observations $\mathbf{D} = \{(x_1, y_1), \cdots, (x_n, y_n)\}$ where $x_n$ are all previous queries with corresponding observations $y_i = f(x_i) + \epsilon$ for a function with noise $\epsilon \in \mathcal{N}(0, \sigma_n^2)$, then the posterior distribution over $f(x)$ is $\mathcal{N}(\mu(x), \sigma^2(x))$ where:

$$\mu(x) = \mathbf{k}(x, \mathbf{X})(\mathbf{k}(\mathbf{X}, \mathbf{X}) + \sigma_n^2 I)^{-1}\mathbf{y} \tag{2}$$

$$\sigma^2(x) = \mathbf{k}(x, x) - \mathbf{k}(\mathbf{X}, x)(\mathbf{k}(\mathbf{X}, \mathbf{X}) + \sigma_n^2 I)^{-1}\mathbf{k}(x, \mathbf{X}) \tag{3}$$

with $\mathbf{X} = (x_1, \cdots, x_n)$ and $\mathbf{y} = (y_1, \cdots, y_n)$.

When $f(x)$ includes time, the observation varies, which requires the GP to model the time change. For DBO (Bogunovic et al., 2016), the $n \times n$ covariance kernel $\mathbf{k}(\mathbf{X}, \mathbf{X}) = \mathbf{k}((x_i, t_i), (x_j, t_j))_{i,j \leq n}$ is adapted to capture the temporal dynamics where $x_i, t_i \in \mathbf{S} \times \mathbf{T}$ for dataset $\mathbf{D} = \{((x_1, t_1), y_1), \cdots, ((x_n, t_n), y_n)\}$. The corresponding observations $y_i = f(x_i, t_i) + \epsilon$ and posterior distribution $f(x, t)$ are dependent with time $\mathcal{N}(\mu(x, t), \sigma^2(x, t))$. This in turn updates $\mu(x, t)$ and $\sigma^2(x, t)$ in equations 2 and 3 where $\mathbf{X} = ((x_1, t_1), \cdots, (x_n, t_n))$ and $\mathbf{K}(x, \mathbf{X}) \rightarrow \mathbf{K}((x, t), \mathbf{X})$.

An acquisition function $\alpha$, determines the next query $x_{n+1}$ for the next iteration of BO by quantifying the advantages of selecting a list queries. In DBO, the next queries are restricted to $t$ as probes cannot occur in the past or future, only the present. Acquisition functions help explore the GP to improve similarity to $f(x)$, in turn exploiting $f(x)$ to find the optimum point. The most common acquisition functions are upper confidence bound (UCB)(Srinivas et al., 2012), expected improvement (EI) (Mockus, 1994), and probability of improvement (PI) (Jones et al., 1998). The next query is found by $x_{n+1} = argmax_{x \in \mathbb{X}} \alpha(x)$ where:

$$UCB(x) = \mu(x, t) + \sqrt{\beta} \cdot \sigma(x, t) \tag{4}$$

$$EI(x) = \mathbb{E}(max(f(x, t) - f^*, 0)) \tag{5}$$

$$PI(x) = P(f(x, t) \geq f^*) \tag{6}$$

with $\beta$ being an exploratory or exploit trade-off variable and $f^*$ being the best observation from $\mathbf{D}$.

Research in DBO models focuses on forgetting-remembering trade-offs where observations further in the past may be regarded as irrelevant with respect to more recent observations. Bogunovic et al.

(2016) introduced two algorithms for these trade-offs, R-GP-UCB which resets the GP at regular intervals based on how much the function has changed and TV-GP-UCB which incorporates data staleness weights to phase queries out over time. Adaptive BO (ABO) by Nyikosa et al. (2018) furthers this work by exploiting temporal correlations of $f(x, t)$ and when to make probes to optimize observations. Chen & Li (2021) builds Transfer BO (TBO) which improves upon ABOs performance by adjusting the GP by augmenting the covariance function by calculating the relationship between the past observations and the current observations. Li et al. (2022) then improves TBO, creating data-driven transfer optimization (DETO) which modifies their previous work. These modifications reduce the hyperparameters in their adjusted GP, an initialization method that does not rely on random initial probes to start DBO, and a new method evolutionary algorithm to adapt UCB to improve performance. However, existing algorithms assume discrete, evenly-spaced time intervals without considering computational response time, which scales as $\mathcal{O}(n^3)$ with dataset size $n$ in BO frameworks. Evenly-spaced intervals are only realistic in online settings where the time to probe and receive observations exceeds the DBO computational cycle time (e.g. robot movement, biofeedback applications).

In practice, DBO response time directly impacts probing frequency, which affects the surrogate model's ability to accurately represent the objective function. Furthermore, current approaches remove observations based solely on their age rather than their relevance to the GP. This temporal filtering discards potentially valuable information, forcing the acquisition function to re-explore previously sampled regions and leading to inefficient observation allocation. Bardou et al. (2024) highlights these issues by designing W-DBO, an observation removal policy for DBO that calculates how relevant an observation is using Wasserstein distance to the current GP and removing any irrelevant observation based on a calculated budget, keeping $n$ small while also calculating relevancy in the least time required. W-DBO improved DBO optimization and was further improved by BOLT (Bardou & Thiran, 2025) by replacing the budget with a specific dataset size calculated by a derived equation.

Although these advancements have improved DBOs ability to locate global optimums across $f(x, t)$, there has been little work to try to design an acquisition function specifically for dynamic environments. In the DBO literature, standard UCB is commonly used as the acquisition function $(x_{n+1}, t_{n+1}) = argmax_{x \in \mathbb{X}} \alpha(x, t)$. This lack of research gives an opportunity of exploration to develop an acquisition function to improve UCB-based DBO model performance. In the next section, we look at AIF and its application as an acquisition function for DBO.

## 2.2 Active Inference

AIF describes optimal behavior that is derived from the need for animals to make actions based on evidence from perception, planning and learning (Friston, 2010). AIF is an emerging theory on brain function (Parr et al., 2020; Friston et al., 2023b; Barp et al., 2022a) and the theory has been extrapolated to machine learning and explanations to human-computer interaction (Murray-Smith et al., 2024; Friston et al., 2024).

AIF oftentimes follows a Markov decision process (MDP) or partially-observed MDP (POMDP) where each hidden state $\mathbf{S} = (s_1, \cdots, s_n)$ receives an observation $\mathbf{O} = (o_1, \cdots, o_n)$ and an action $u_n$ can be performed at transfer to a new hidden state $s_{n+1}$ to receive a new observation $o_{n+1}$. As opposed to reinforcement learning in machine learning which adjusts policy $\pi = (u_1, \ldots, u_n)$ to maximize reward, AIF minimizes variational free energy $F(\pi)$ which is a measure of uncertainty in the next observation. $F(\pi)$ can be calculated as an evidence lower bound:

$$F(\pi) = D_{KL}[Q(\mathbf{S} \mid \pi) \parallel P(\mathbf{S} \mid \mathbf{O}, \pi)] - lnP(\mathbf{O} \mid \pi) \tag{7}$$

with $D_{KL}[Q(\mathbf{S} \mid \pi) \parallel P(\mathbf{S} \mid \mathbf{O}, \pi)]$ representing the Kullback-Leibler divergence between the approximate posterior belief $Q(\mathbf{S} \mid \pi)$ and the true posterior $P(\mathbf{S} \mid \mathbf{O}, \pi)$, and $lnP(\mathbf{O} \mid \pi)$ representing the likelihood $o_n$ will occur under $\pi$. $F(\pi)$ is minimal when the approximate posterior identically replicates the true posterior and when there is no uncertainty that an observation will occur for the current policy.

AIF agents minimize expected free energy by selecting policies that reduce uncertainty about future states. The generative model for each policy requires three components: a likelihood matrix $P(\mathbf{O} \mid \mathbf{S})$ mapping states to observations, a transition matrix $P(s_t \mid s_{t-1}, u_{t-1})$ modeling state dynamics, and an outcome prior $P(\tilde{o})$ representing the probability distribution over desired observations. The

optimal policy $\pi^*$ corresponds to the minimum expected free energy $G(\pi^*)$ where:

$$G(\pi) = \mathbb{E}_{\mathbf{Q}}[lnQ(\mathbf{S} \mid \pi) - lnP(\mathbf{O}, \mathbf{S})] \tag{8}$$

$$\approx -\mathbb{E}_{Q(\mathbf{O}|\pi)}[D_{KL}[Q(\mathbf{S} \mid \mathbf{O}) \parallel Q(\mathbf{S} \mid \pi)]] - \mathbb{E}_{\mathbf{Q}(\mathbf{O},\mathbf{S}|\pi)}[lnP(\tilde{o})] \tag{9}$$

with equation 8 compares the agents expectation with what is probable in the given generative model. In equation 9, $D_{KL}[Q(\mathbf{S} \mid \mathbf{O}) \parallel Q(\mathbf{S} \mid \pi)]$ represents the intrinsic value (information gain) expected from the current policy and visiting the specific state and $\mathbb{E}_{\mathbf{Q}}[lnP(\tilde{o})]$ represents the extrinsic value (goal satisfaction) of how likely the current policy will achieve the desired outcomes which equates to $\mathbf{C}$ (Barp et al., 2022b). As we aim to minimize $G(\pi)$ and both terms are negative, the agent therefore maximizes the expected information gain and probability of achieving the desired state. However, Millidge et al. (2021) showed that $G(\pi)$ from equation 9 is not the only formulation of expected free energy. They proposed an alternative function that focuses on minimizing the free energy of the expected future based on free energy principles:

$$G(\pi) = -\mathbb{E}_{Q(\mathbf{O}|\pi)}[D_{KL}[Q(\mathbf{S} \mid \mathbf{O}) \parallel Q(\mathbf{S} \mid \pi)]] + \mathbb{E}_{\mathbf{Q}(\mathbf{O},\mathbf{S}|\pi)}[D_{KL}[Q(\mathbf{O} \mid \mathbf{S}) \parallel P(\tilde{o})]] \tag{10}$$

Equation 10 maintains the same intrinsic value (first term; the expected information gain), but replaces the extrinsic component. Instead of maximizing expected evidence for desired observations, the free energy of expected futures minimizes the KL-divergence between the generative model's predicted observation likelihood and the preferred observation distribution $P(\tilde{\mathbf{O}})$. This formulation encourages the agent to select actions leading to observations close to preferred outcomes while simultaneously seeking states with high observation entropy; effectively exploring regions where the generative model is uncertain. This adaptation is particularly suited to environments where the agent lacks knowledge of the true generative model and its temporal evolution, making it well-suited for DBO applications where the objective function dynamics are unknown.

In DBO, queries at time cannot access future observations. An acquisition function selects the next queries based on specific criteria (e.g. equation 4) that tries to explore and exploit the surrogate model. In this paper, we use (a) fixed-observation budget and (b) fixed run-time where faster acquisition allows more queries for evaluation to show different tradeoffs between surrogate performance and acquisition cost. In the following section, we determine the application of an AIF acquisition function as a proof of concept to understand how effective free energy principles will be at discovering optimal probes through time.

## 3   ACTIVE INFERENCE FOR DYNAMIC BAYESIAN OPTIMIZATION

In this section we propose BOBA, an acquisition function that utilizes free energy principles to decide the next input for DBO. The reasoning on why AIF would be suitable as an acquisition function was inspired by the frameworks ability to maximize reward. Da Costa et al. (2023) demonstrated on a Markov decision processes (MDP) in a finite horizon, active inference can produce policies that are Bellman optimal.

However, Da Costa et al. (2023) demonstrated Bellman optimality in MDP environments that do not require further exploration (e.g. as they are time invariant) and where the transition matrix is fully known. Since DBO requires real-time learning with unknown dynamics, the standard AIF formulations must be modified for an AIF-based acquisition function to effectively discover optima with an unknown generative model (Parr et al., 2020).

While AIF was initially applied to simple environments like grid worlds or T-maze problems with easily modeled distributions and outcomes, the complexity of the required generative model scales with environmental complexity. In DBO settings, this challenge is compounded because the generative model itself evolves over time in ways unknown to the agent. As a MDP requires discrete states and observations, the GP is converted to a discrete space to estimates the generative model at time $t$ where $\mathbf{S_t} = ((x_1, t), \cdots, (x_n, t))$ and $\mathbf{O_t} = (y_1, \cdots, y_n)$ where

**Assumption 1:** $\mathbf{S_t} = disc(\mathbf{GP_t})$ where $\mathbf{GP_t} \in \mathbb{R}^n$ and $n = 262144^{\frac{1}{d-1}}$. $n$ represents the number of discrete inputs for each dimension of the GP excluding time using deterministic sampling. 262144 was decided on the basis of the compromise that a value too small will limit the options for $\mathbf{S_t}$ to select and too large will increase the computation time, decreasing the speed of the BOBA model while also giving an integer value for all selected dimensions. We partition the continuous $GP_t$ into

a finite number of regions $disc(\mathbf{GP_t})$ and assign discrete state labels accordingly to mimic an MDP environment.

Additionally, as DBO can select any input irrelevant to the previous input which are not Markovian, $(x_n, t) = u_n$. This allows BOBA to select whichever policy will minimize free energy where $\pi = \mathbf{S_t}$, as the policy is selected with a limit of one input. Although the optimal observation for each time point is independent of historical observations, the posterior belief $Q(\mathbf{S_{t+1}} \mid \mathbf{O_t})$ depends on all previous observations and state combinations currently training the GP. This dependency propagates through to the expected free energy $G(\pi)$, requiring adaptation for the application of DBO setting.

In DBO, actions in AIF can represent a sequence determined on the policy. Unfortunately, DBO has no knowledge of the upcoming time-steps, so creating a policy until the final observation $o_n$ would be infeasible, especially since $f(x, t)$ varies with time. As policies based on sequences would not be feasible without transfer learning, $\pi$ only describes how the next query at time-step $t$ is selected based on minimizing $G(\pi)$.

To create a generative model that adapts with time, we use the GP estimation at time $t$ to estimate the extrinsic and intrinsic values from equation 10. The GP allows calculation of $Q(\mathbf{O_t} \mid \mathbf{S_t})$ and prior knowledge will select $P(\tilde{o})$ with the following assumptions

**Assumption 2:** $\mathbf{O}_t = \mathcal{N}(\mu(\mathbf{S_t}), \sigma^2(\mathbf{S_t}))$ is based on $\mathbf{GP_t}$. The GP is normalized where $\mathbf{GP_t} \in \mathcal{N}(0, 1)$.

As AIF is applied to discrete environments as opposed to continuous, we need to have discrete inputs (equivalent to states) for the GP so the AIF formula can function at a low computational cost. Assumption 2 relies on the surrogate model's ability to accurately model observations. If GP cannot accurately model the function and underfit the posterior, it can be assumed performance of the BOBA function will deteriorate. As large dimension functions will require more states to cover the full input space of the GP, $\mathbf{S}$ is batched to improve speed of calculating $\mathbf{O_t}$. The desired observation $P(\tilde{o})$ is selected with prior knowledge of the tested functions global optima which is stated in the Appendix. As the global optima will not be attainable at every time-point, the local optima at each time-step will be sought after based on equation 10 which was utilized for BOBA. However, adaptations are required due to limitations of the GP.

$$G(\pi) \approx -\mathbb{E}_{Q(\mathbf{S_{t+1}}, \mathbf{O_t}|\pi)}[D_{KL}[Q(\mathbf{S_{t+1}} \mid \mathbf{O_t}) \parallel Q(\mathbf{S_t})]] + \mathbb{E}_{\mathbf{Q(O_t, S_t}|\pi)}[D_{KL}[Q(\mathbf{O_t} \mid \mathbf{S_t}) \parallel P(\tilde{o})]]$$
(11)

The first term on the right side represents the intrinsic value; in terms of the GP $Q(\mathbf{S_t})$ represents the prior and $Q(\mathbf{S_{t+1}} \mid \mathbf{O_t})$ represents the posterior with this term maximizing the change in the GP once observation $\mathbf{O_t}$ has occurred. Unfortunately, to calculate how much the GP will adapt for each observation would require backward induction to calculate which is infeasible for DBO which depends on computational timing. For this reason, we use uncertainty sampling (Krause & Hübotter, 2025) to maximize mutual information for the intrinsic value:

$$\mathbb{E}_{Q(\mathbf{O_t}|\pi)}[D_{KL}[Q(\mathbf{S_{t+1}} \mid \mathbf{O_t}) \parallel Q(\mathbf{S_t})]] \approx \frac{1}{2}\log(1 + \frac{\sigma^2(\mathbf{S_t})}{\sigma_n^2})$$
(12)

where $\sigma^2(\mathbf{S_t})$ represents the standard deviance at at all discrete states and $\sigma_n^2$ represents the epistemic noise from the GP. The formulation directly targets reducing uncertainty which aligns with the left side of equation 12 with maximizing information seeking and updating the GP while also maintaining computational efficiency. Equation 12 represents a trade-off between theoretical optimality and practical feasibility. While uncertainty sampling may not capture an accurate measurement of information seeking, it provides a computationally tractable method that maintains the exploratory benefits essential for dynamic environments. Additionally, this formulation comes with the assumption:

**Assumption 3:** The input is homoscedastic noise as if there are inputs with large aleatoric uncertainty with respect to epistemic uncertainty, equation 12 will solely target these inputs.

A limitation of utilizing a GP as the surrogate model is the estimated uncertainty in unexplored regions of $f(x, t)$ are equal which may be false in real-world settings, limiting BOBA's generative model understanding and urgency to explore. For this assumption, we focus on simulations with fixed noise across inputs.

The second term on the right side of equation 11 represents the divergence between the expected outcome of inputs $\mathbf{S_t}$ and the selected desired observations which leads the selected state to find observations similar to $P(\tilde{o})$. This formulation equates to the following equation:

$$\mathbb{E}_{\mathbf{Q}(\mathbf{O_t}, \mathbf{S_t}|\pi)}[D_{KL}[Q(\mathbf{O_t} \mid \mathbf{S_t}) \parallel P(\tilde{o})]] \approx D_{KL}(\mathcal{N}(\mu(\mathbf{S_t}), \sigma^2(\mathbf{S_t})) \parallel \mathcal{N}(\tilde{o}, \sigma_n^2)) \tag{13}$$

$$\approx \log(\frac{\sigma_n}{\sigma(\mathbf{S_t})}) + \frac{\sigma^2(\mathbf{S_t}) + (\mu(\mathbf{S_t}) - \tilde{o})^2}{2\sigma_n^2} - \frac{1}{2} \tag{14}$$

where $\mu(\mathbf{S_t}), \sigma^2(\mathbf{S_t})$ are the discrete outputs of the GP in terms of mean and standard deviation. By minimizing equation 14, our policy will select inputs that are estimated to achieve the desired observation.

To adjust BOBA's exploration rate due to assumption 3, we multiply the intrinsic value from equation 12 with a $\beta$ value to adjusts BOBA's exploration ability similar to UCB equation 4. Additionally, as the extrinsic values $\in [0, \infty]$ and the intrinsic value range are significantly smaller, this may lead to the extrinsic value overpowering the intrinsic value, leading to searching only local optima. To evaluate multiple approaches, we normalize both the extrinsic value and the intrinsic value using min-max normalization to scale outputs $\in [0, 1]$. The BOBA function that uses the normalized values is referred to as BOBA-N whereas the standard values are used in the standard BOBA model. Once $G(\pi)$ for all possible inputs are calculated the following equation identified the next input the DBO model will select:

$$x_t = argmax(softmax(-G(\pi))) \tag{15}$$

This process repeats each time the BOBA acquisition function is called in DBO. Pseudo-code for the BOBA function is provided below:

---
**Algorithm 1** BOBA

---
$\mathbf{GP_t}$, preferred observation $\tilde{o}$
**if** DBO requests $x_t$ **then**
    Calculate $\mathbf{S_t}$ based on a fixed number $n$
    Calculate $\mathbf{O}_t = \mathcal{N}(\mu(\mathbf{S_t}), \sigma^2(\mathbf{S_t}))$ for all possibilities of $\mathbf{S_t}$
    Calculate $G(\pi)$ for all $\mathbf{S_t}$ combinations
    Provide $x_t = max(softmax(-G(\pi)))$ for the DBO system
**end if**

---

## 4 NUMERICAL RESULTS

In this section, we measure the performance of the BOBA function through hyperparameter tuning and benchmarked against other models. We use mean regret for evaluation, the difference between all observations and the best observation at their respective time step and simulation function.

BOBA was evaluated using eight synthetic benchmarks which are replicated in Bardou & Thiran (2025) paper's evaluation. All dimensions $\mathbf{S_t}$ were normalized between $[0, 1]^d$ where each benchmark had specific bounds stated in section A.2. To evaluate all models, we consider two realistic scenarios that reflect different computational bottlenecks in dynamic optimization.

**Fixed observations** limits each model to 120 queries over a 10-minute horizon, simulating expensive black-box functions requiring approximately 5 seconds per evaluation. This 5-second evaluation time reflects real-world optimization applications such as biofeedback systems (Rodriguez-Larios & Alaerts, 2021; Afergan et al., 2014), or training for data transfer tasks (Swargo et al., 2025) where function evaluation dominates computational cost. The 120-query limit balances two competing requirements: providing sufficient observations for models to learn dynamic patterns while maintaining realistic constraints where function changes occur faster than comprehensive exploration allows. Although this evaluation protocol is uncommon in existing DBO literature where models are typically evaluated without observation constraints, it represents a critical practical scenario where the black-box system's computational expense exceeds the surrogate model's query selection time.

**Fixed time** alternatively constrains each model to 10 minutes of wall-clock time, allowing faster algorithms to make more queries. This scenario reflects applications where black-box evaluations

are rapid but optimization decisions must be made quickly, and has been validated in prior work (Bardou et al., 2024; Bardou & Thiran, 2025).

In order to mimic the variability found in real-world applications, we added random noise to the synthetic benchmarks. Specifically, we incorporated Gaussian noise equal to 2.5% of each objective function's value.

For benchmarking purposes we used GP-UCB and WDBO (Bardou et al., 2024) with more information on their performance stated in section A.3. The reason further models from section 2.1 were not incorporated was due to a lack of open source code [1].

All models were implemented with the same version of BOTorch and Python 3.12. All models were validated ten times each using an NVIDIA GeForce RTX 3090 GPU.

### 4.1 BOBA HYPERPARAMETER TUNING

By adjusting the $\beta$ value stated earlier, each BOBA implementation will adjust the effect the intrinsic value will have on each simulation. Some simulations would require a larger intrinsic value to increase exploration but too much exploration would lead to an increase of regret due to a lack of exploitation for the current time step.

In section A.1, all figures presented demonstrate how adjusting the $\beta$ affects the regret of each simulation. For standard BOBA and normalized BOBA functions, $\beta = 2^n$ and $\beta = 2^{-n}$ respectively where $n \in [0, 5]$. Results show that some simulations such as Shekel test have very minimal changes with varying $\beta$ values irrespective of the BOBA model. However, tests such as Powell, Ackley and Griewank's average normalized regret changes significantly with $\beta$ and the model used. Both Ackley and Griewank are shown in Figure 1. A more detailed analysis on how and why $\beta$ values affect model performance is presented in section A.1.

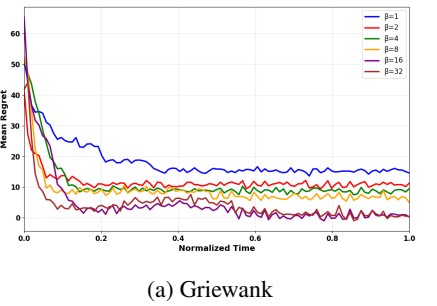
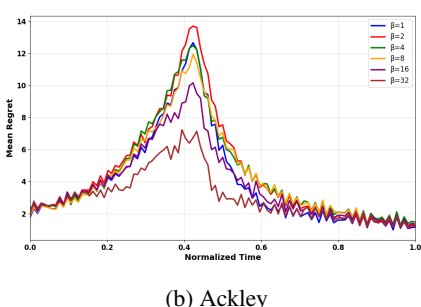

|  (a) Griewank | (b) Ackley |

Figure 1: Mean regret over time for two different simulations using GP-BOBA with fixed observations. Each plot represents a different $\beta$ value as shown in the legend where a larger $\beta$ value represents an increase in exploration.

### 4.2 EXPERIMENTAL SETTING

To benchmark each BOBA model with their respective models, we used the corresponding average regret from the best recorded $\beta$ value.

As demonstrated from Table 1, by adding BOBA to the DBO models the average regret improved significantly except for simulation Shekel and Hartman-3. Shekel's best performing model (GP-UCB) did not perform significantly better compared to the other models, whereas all BOBA consistently performed worse in Hartmann-3.

However, when the fixed horizon is set to time as opposed to observations, the results change significantly where only half of all simulations are performed best with the BOBA models. Additionally, it was unexpected that WDBO would be outperformed by the GP-UCB model even though past papers

---

[1]To follow the trend of open source code, all code used for this paper is located at https/GitHub/Anonymous currently for submission

Table 1: Benchmark of different BOBA combinations with different dynamic bayesian optimization models with a fixed observation limit of 120. (N) represents if extrinsic and intrinsic values are normalized. All BOBA values represent the best regret for different $\beta$ values. Mean regret over 10 independent replications is reported. Bold values show best performance; underlined values show no significant difference from best.

| Experiment (d) | GP-UCB | WDBO | GP-BOBA | GP-BOBA(N) | WDBO-BOBA | WDBO-BOBA(N) |
|---|---|---|---|---|---|---|
| Schwefel (4) | $788 \pm 112$ | $846 \pm 253$ | $\underline{596 \pm 182}$ | $\mathbf{466 \pm 195}$ | $699 \pm 232$ | $\underline{589 \pm 233}$ |
| Powell (4) | $903 \pm 249$ | $2025 \pm 673$ | $\mathbf{452 \pm 245}$ | $\underline{557 \pm 152}$ | $782 \pm 244$ | $1532 \pm 462$ |
| Eggholder (2) | $\underline{351 \pm 47}$ | $506 \pm 54$ | $372 \pm 17$ | $\mathbf{333 \pm 44}$ | $517 \pm 45$ | $510 \pm 47$ |
| Ackley (4) | $\underline{3.92 \pm 1.25}$ | $4.57 \pm 1.68$ | $3.08 \pm 1.29$ | $\mathbf{3.05 \pm 1.30}$ | $3.39 \pm 1.33$ | $4.27 \pm 1.87$ |
| Shekel (4) | $\mathbf{1.62 \pm 0.13}$ | $\underline{1.72 \pm 0.12}$ | $1.86 \pm 0.19$ | $\underline{1.68 \pm 0.13}$ | $1.99 \pm 0.14$ | $\underline{1.71 \pm 0.07}$ |
| Griewank (6) | $\underline{4.55 \pm 0.84}$ | $59.68 \pm 23.80$ | $\mathbf{4.29 \pm 0.98}$ | $10.95 \pm 0.87$ | $12.87 \pm 4.37$ | $15.44 \pm 8.24$ |
| Hartmann3 (3) | $\mathbf{0.21 \pm 0.03}$ | $0.26 \pm 0.06$ | $0.53 \pm 0.08$ | $0.48 \pm 0.10$ | $0.69 \pm 0.38$ | $0.49 \pm 0.15$ |
| Hartmann6 (6) | $1.05 \pm 0.14$ | $1.02 \pm 0.21$ | $1.70 \pm 0.27$ | $\mathbf{0.55 \pm 0.05}$ | $1.69 \pm 0.31$ | $\underline{0.64 \pm 0.13}$ |

have shown different results (Bardou et al., 2024; Bardou & Thiran, 2025). Further discussion on this point is continued in section A.3. As shown in Figure 2, when observations as fixed as opposed

Table 2: Benchmark of different BOBA combinations with different dynamic bayesian optimization models with a fixed time limit of 10 minutes. (N) represents if extrinsic and intrinsic values are normalized. All BOBA values represent the best regret for different $\beta$ values. Mean regret over 10 independent replications is reported. Bold values show best performance; underlined values show no significant difference from best.

| Experiment (d) | GP-UCB | WDBO | GP-BOBA | GP-BOBA(N) | WDBO-BOBA | WDBO-BOBA(N) |
|---|---|---|---|---|---|---|
| Schwefel (4) | $817 \pm 133$ | $884 \pm 188$ | $\mathbf{261 \pm 159}$ | $\underline{305 \pm 153}$ | $510 \pm 253$ | $561 \pm 180$ |
| Powell (4) | $\underline{239 \pm 47}$ | $1493 \pm 894$ | $\mathbf{228 \pm 26}$ | $1161 \pm 251$ | $557 \pm 209$ | $1225 \pm 343$ |
| Eggholder (2) | $\mathbf{175 \pm 24}$ | $427 \pm 52$ | $255 \pm 55$ | $221 \pm 38$ | $420 \pm 90$ | $389 \pm 81$ |
| Ackley (4) | $\underline{3.00 \pm 1.24}$ | $4.02 \pm 2.12$ | $3.36 \pm 1.23$ | $\mathbf{2.68 \pm 1.73}$ | $5.23 \pm 1.61$ | $4.65 \pm 2.28$ |
| Shekel (4) | $\mathbf{1.58 \pm 0.16}$ | $1.83 \pm 0.13$ | $2.02 \pm 0.14$ | $\underline{1.61 \pm 0.09}$ | $1.96 \pm 0.15$ | $1.74 \pm 0.08$ |
| Griewank (6) | $\mathbf{1.79 \pm 0.23}$ | $14.04 \pm 8.46$ | $2.04 \pm 0.20$ | $6.51 \pm 0.78$ | $4.63 \pm 1.76$ | $10.4 \pm 5.59$ |
| Hartmann3 (3) | $\mathbf{0.09 \pm 0.01}$ | $0.14 \pm 0.05$ | $0.34 \pm 0.05$ | $0.35 \pm 0.08$ | $0.36 \pm 0.10$ | $0.38 \pm 0.09$ |
| Hartmann6 (6) | $0.43 \pm 0.03$ | $0.61 \pm 0.07$ | $1.34 \pm 0.24$ | $\mathbf{0.39 \pm 0.11}$ | $1.57 \pm 0.22$ | $0.65 \pm 0.47$ |

to time, GP-BOBA significantly outperformed GP-UCB. However, when time is fixed to 10 minutes both GP-BOBA and UCB have similar performances. This is shown in Table 2 where GP-UCB performs similar to GP-BOBA and/or GP-BOBA (N) when the simulations does not change as drastically with time, but when the simulations change rapidly as shown in Table 1, GP-BOBA models significantly outperform GP-UCB.

This finding can be reinforced by WDBO's performance in comparison to WDBO-BOBA and WDBO-BOBA (N) and how they alternate between fixed time or observation horizon. In Table 2, WDBO has significantly worse performance compared to WDBO-BOBA and WDBO-BOBA (N) in 3 tests (Schwefel, Powell, and Griewank). In Table 1, WDBO performance in comparison to both WDBO-BOBA models significantly worsens, where WDBO-BOBA and BOBA (N) not only achieve significantly different in two additional tests (Ackley and Hartmann6) but also achieve results similar to the best benchmarks for each simulation. These findings show a BOBA acquisition function has the ability to enhance DBO models in environments where the black box function are computationally costly and require time greater than it takes for the DBO model to complete an iteration while slightly improving performance when the black box function is not computationally costly and the model computational time matters.

Based on both tables, there is some variability as to whether BOBA or BOBA (N) functions will improve performance for specific simulations. Both the Griewank and Hartmann6 regret with time for each model are shown in Figure 3 to demonstrate why one model performs better in different situations. In Figure 3a, GP-BOBA minimizes regret and maintains an average regret level whereas GP-BOBA (N) increases regret in the middle of the function. This is as Griewank resembles a bowl shape (equation can be found at equation A.2.6) where further exploration may result in unnecessary accumulation of regret. Alternatively, the opposite is true for Hartmann6 (As shown in equation A.2.8) where exploration is required as the function adapts significantly with time. As shown in

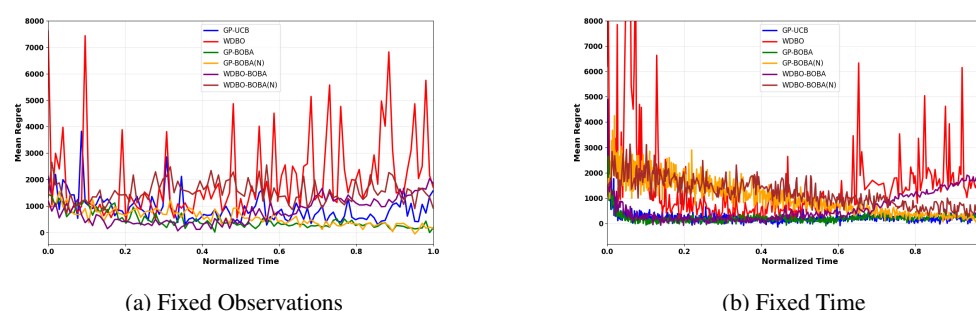

(a) Fixed Observations                    (b) Fixed Time

Figure 2: Mean regret over time for the Powell simulation with different fixed horizons. Each plot represents a different DBO model as shown in the legend.

Figure 3b, regret increases for all models around normalized time 0.2. GP-BOBA (N) performs the best as it is able to explore after accumalating regret that starts to decrease, whereas GP-BOBA repeats the same error due to its limited ability to explore.

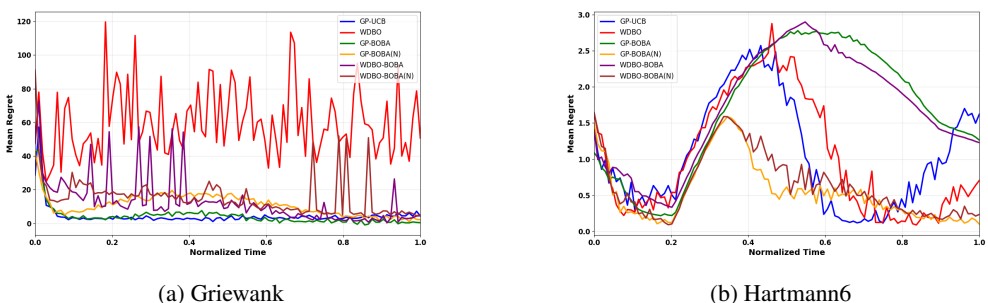

(a) Griewank                             (b) Hartmann6

Figure 3: Mean regret over time for two simulation with fixed observations. Each plot represents a different DBO model as shown in the legend. Figure (a) shows that the BOBA function is better suited when no extensive exploration is required whereas Figure (b) shows that the BOBA (N) function is better suited when obstacles require extensive exploration.

## 5    CONCLUSION

In this paper, we demonstrated how and why Active Inference (AIF) can improve the performance of Dyanmic Bayesian Optimization (DBO) depending if the fixed horizon is based on the number of observations or the time of the session. Additionally, BOBA demonstrates how surrogate models can produce a generative model in real time, allowing AIF researchers a new method to model an agents dynamic environment.

BOBA was identified to enhance DBO models when the black box function is computationally costly as shown by BOBA's performance in fixed observation horizon settings. However, the BOBA model performance was dependent on several factors such as the Gaussian Process (GP) reliably modeling the true function. Even though this work does not look into different kernel and kernel parameters, it can be assumed that choosing a kernel that is better suited to accurately model each function would improve BOBA's performance as $Q(\mathbf{O_t} \mid \mathbf{S_t}) \rightarrow P(\mathbf{O_t} \mid \mathbf{S_t})$. In future works, we will determine how different surrogate models will affect BOBA's performance while minimizing hyperparameter tuning.

Additionally, as the original formulation of intrinsic value would require a significant amount of time to compute for continuous space, BOBA estimates information gain by maximizing mutual information. As BOBA is a proof of concept, there may be formulation that is applicable for GP surrogate models which could in turn improve BOBA's performance in future works.

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

# A    APPENDIX

## A.1    BOBA $\beta$ PARAMETER TUNING

This section analyzes how the exploration-exploitation parameter $\beta$ affects BOBA performance across different test functions. Standard BOBA increases exploration with higher $\beta$, while normalized BOBA(N) achieves maximum exploration at $\beta = 1$. All results are shown in the Figures below which gives further detail on how performance varied across models, functions, and $\beta$ values.

**Shwefel and Griewank:** How the $\beta$ value affected performance for these functions depended on if the BOBA function was standard or normalized. For models that utilize the standard BOBA function, performance was best when $\beta = 32$, suggesting an increase in exploration improves regret minimization. However, the BOBA (N) function minimized regret effectively when exploration is minimized. This dichotomy exists as the standard BOBA highlights exploitation where BOBA (N) was made to drastically increase exploration so both acquisition functions could have a use case for different simulations.

These findings show that the Schwefel and Griewank functions required a balance of the $\beta$ hyperparameter between exploration and exploitation. This is inferred due to both functions requiring a balance between exploration and exploitation due to the many local optima's present in their sinusoidal landscape.

**Powell:** When the BOBA (N) function is selected, regret is minimized when exploration is minimized due to exploration increasing regret for this specific simulation. For the GP-BOBA function, performance is best when $\beta = 16$ as exploration improves the model's abilities to locate global optima as opposed to staying in local optima. However, with these specific models, regret drastically increases if exploration increases further, demonstrating the findings shown with the BOBA (N) models requiring minimized exploration. Additionally, a similar relationship is found for the WDBO-BOBA (N) models, but their relationship with performance and their $\beta$ value differs between fixed observation and time horizons. It is unclear why, but the authors assume this is due to how WDBO performs in different fixed horizon settings adjusts drastically the GP which in turn would affect BOBA's performance.

**Eggholder and Shekel:** For these functions, all performance is similar and the majority of models appear independent to $\beta$ changes. There are trends unique to each model but there is no overall trend that explains why specific $\beta$ values perform better than other. As can be seen by Figure 5c and Figure 5e, all $\beta$ values follow a similar pattern which is replicated by all other models evaluated

on the Eggholder function. We assume that this is due to the nature of the function having extreme changes of optimum with time.

However, for the Shekel function in fixed time horizon settings, the $\beta$ value affected performance due to unrestricted number of observations to identify the many optima across the function. This improvement in performance was shown by the BOBA (N) models, where an increase in accumulated regret when $\beta = 1/32$, suggesting that an increase in exploration is required to optimize this function which is validated based on the performances shown in Table 2 where models that explored more performed the best.

**Ackley:** It appears that any model that identifies the global optima early significantly minimize the mean regret which occurs in the fixed observation horizon for the standard BOBA models and GP-BOBA (N) when the $\beta$ value is maximized. For the fixed time horizon, there does not appear to be a clear trend for the BOBA models as some benefit from a greater $\beta$ value such as GP-BOBA and WDBO-BOBA (N) while the GP-BOBA (N) and WDBO-BOBA performs best with smaller $\beta$ values.

**Hartmann 3 and 6:** Both of these functions required extensive exploration compared to the other functions. In terms of which $\beta$ value, these functions have dramatic changes compared to other functions and vary due to what section of each function BOBA is trying to optimize.

For Hartmann3 BOBA models accumulate significant amount of regret in the first half of the simulation, whereas these models find global optima in the second half of the simulation. However, for the BOBA (N) function when $\beta = 1$, this relationship is reversed where the model performs better in the first half of the Hartmann3 function and not the second half.

This is due to the middle section of the function requiring extensive exploration in regions the BOBA function has yet to explore, whereas other models such as the GP-UCB find easier to locate as the BOBA models stronger dependence on the GP. This relationship can be shown in Figure 19g.

For the Hartmann6 function, BOBA (N) shows a trend where a $\beta$ value around 1/8-1/16 is optimal, requiring exploration but at a reduced rate. This trend is not replicated in the standard BOBA models as it appears both Hartmann functions require extensive exploration to minimize regret effectively.

### A.1.1 GP-BOBA (N) FIXED OBSERVATIONS

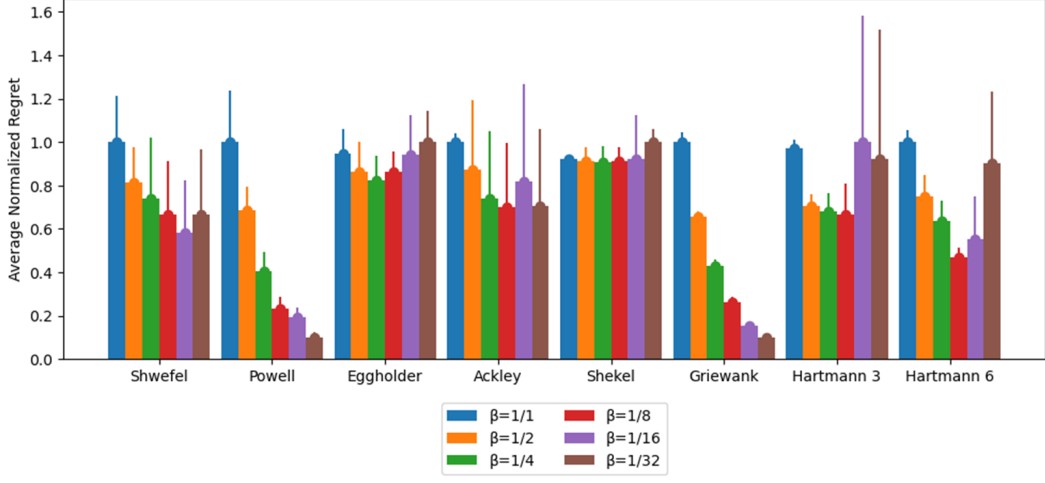

Figure 4: A grouped bar chart showing the variance $\beta$ has on the GP-BOBA (N) model with fixed observation horizon for all simulations

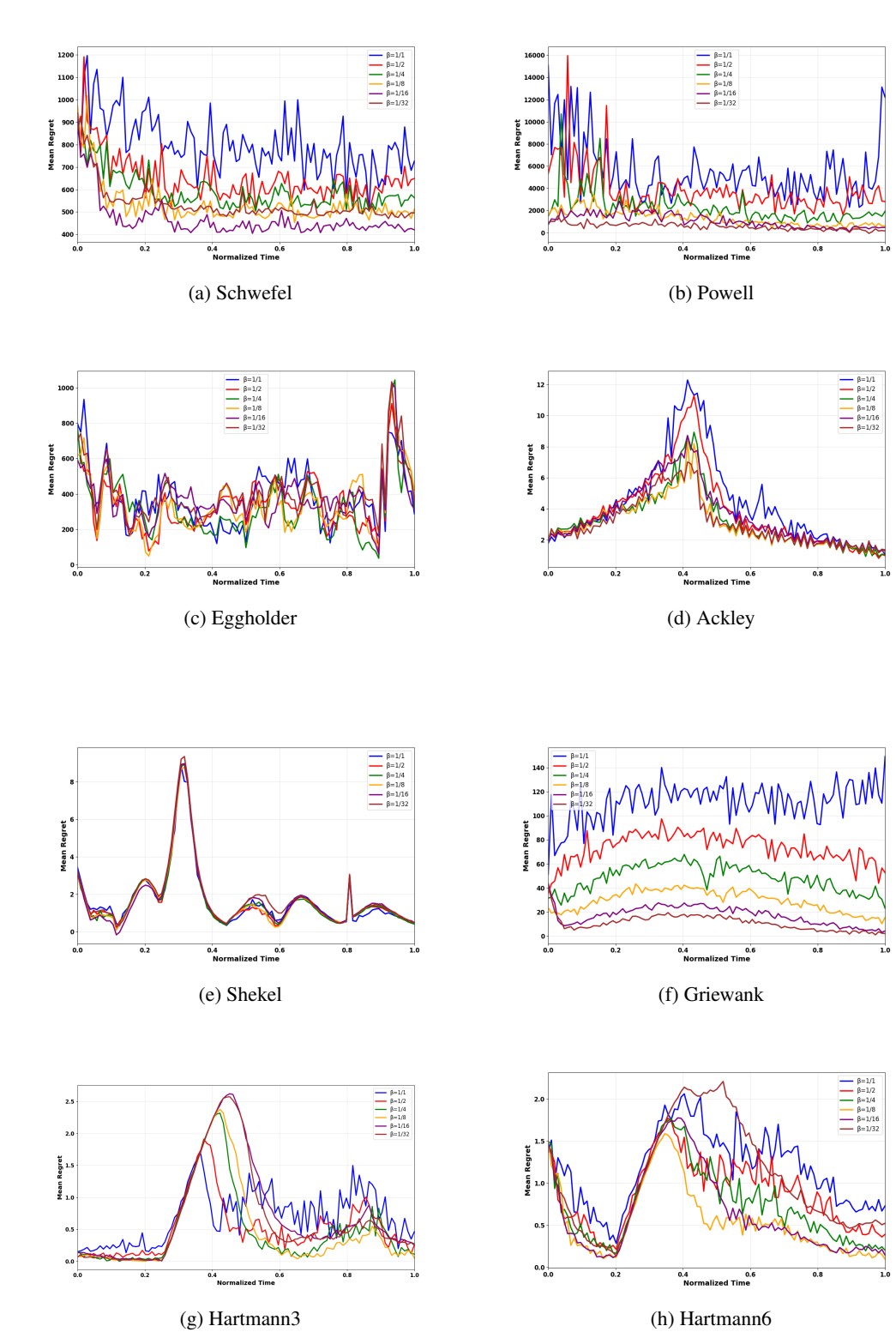

Figure 5: All eight figures show the mean regret with time for all eight simulations for the GP-BOBA (N) model with a fixed observation horizon. All figures show how adapting the $\beta$ value changes the models ability to minimize regret.

### A.1.2 GP-BOBA FIXED OBSERVATIONS

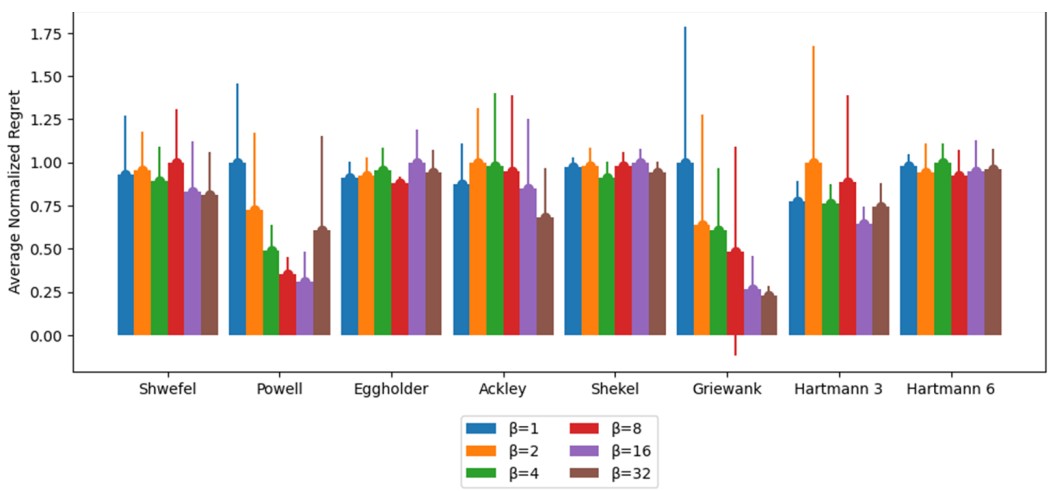

Figure 6: A grouped bar chart showing the variance $\beta$ has on the GP-BOBA model with fixed observation horizon for all simulations

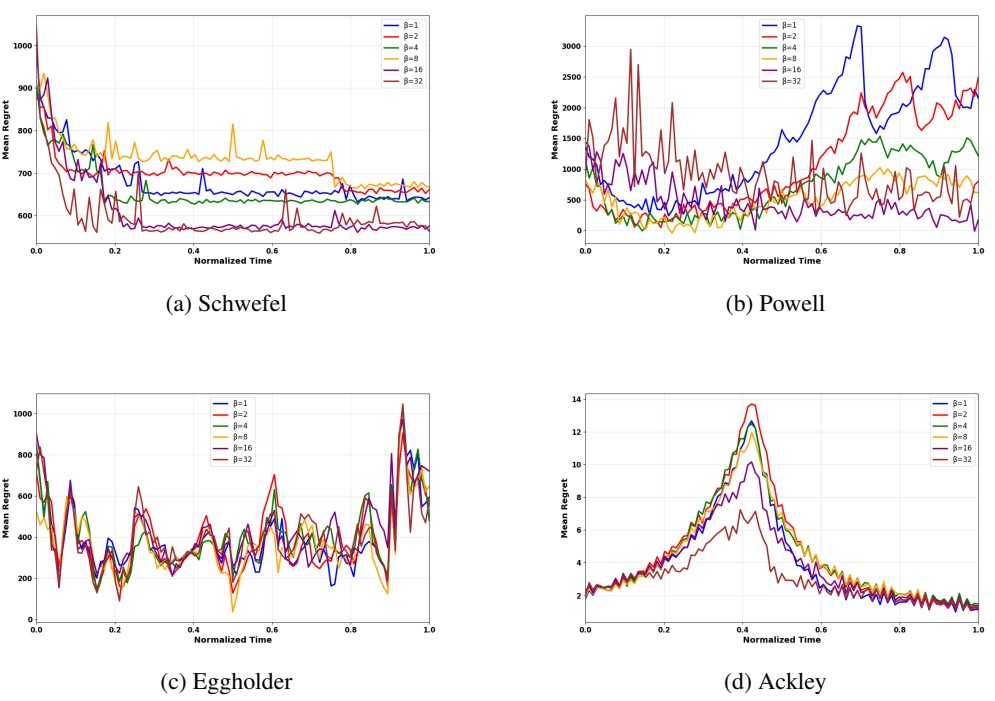

(a) Schwefel

(b) Powell

(c) Eggholder

(d) Ackley

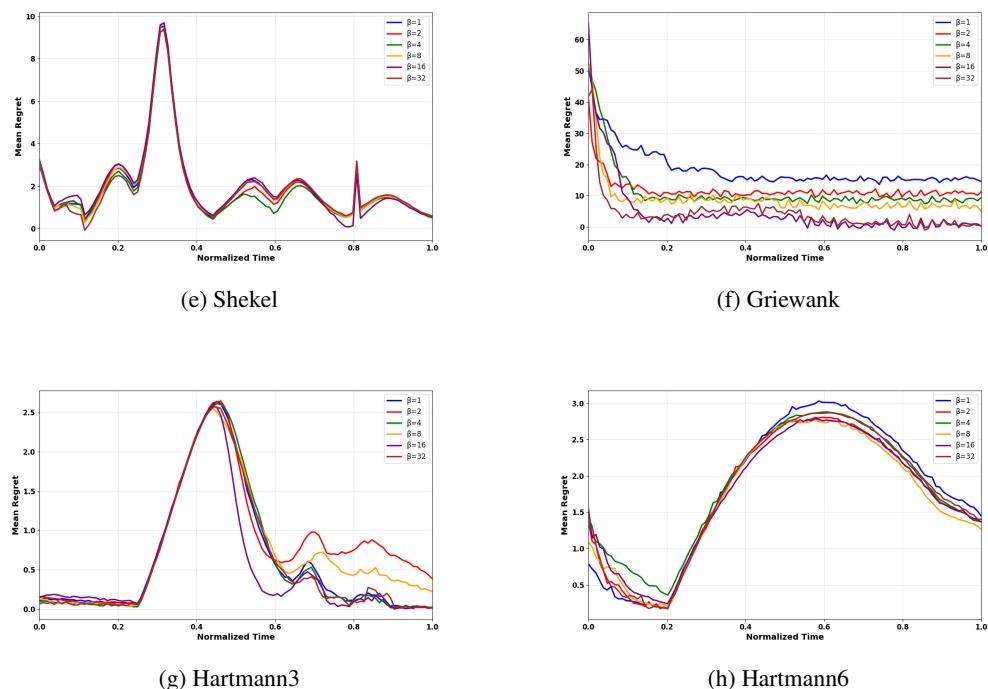

(e) Shekel

(f) Griewank

(g) Hartmann3

(h) Hartmann6

Figure 7: All eight figures show the mean regret with time for all eight simulations for the GP-BOBA model with a fixed observation horizon. All figures show how adapting the $\beta$ value changes the models ability to minimize regret.

### A.1.3 WDBO-BOBA FIXED OBSERVATIONS

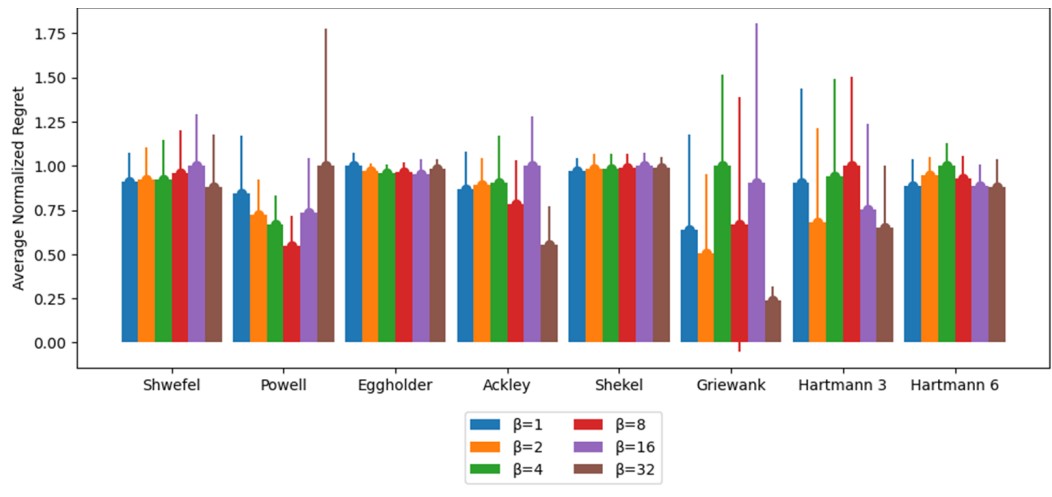

Figure 8: A grouped bar chart showing the variance $\beta$ has on the WDBO-BOBA model with fixed observation horizon for all simulations

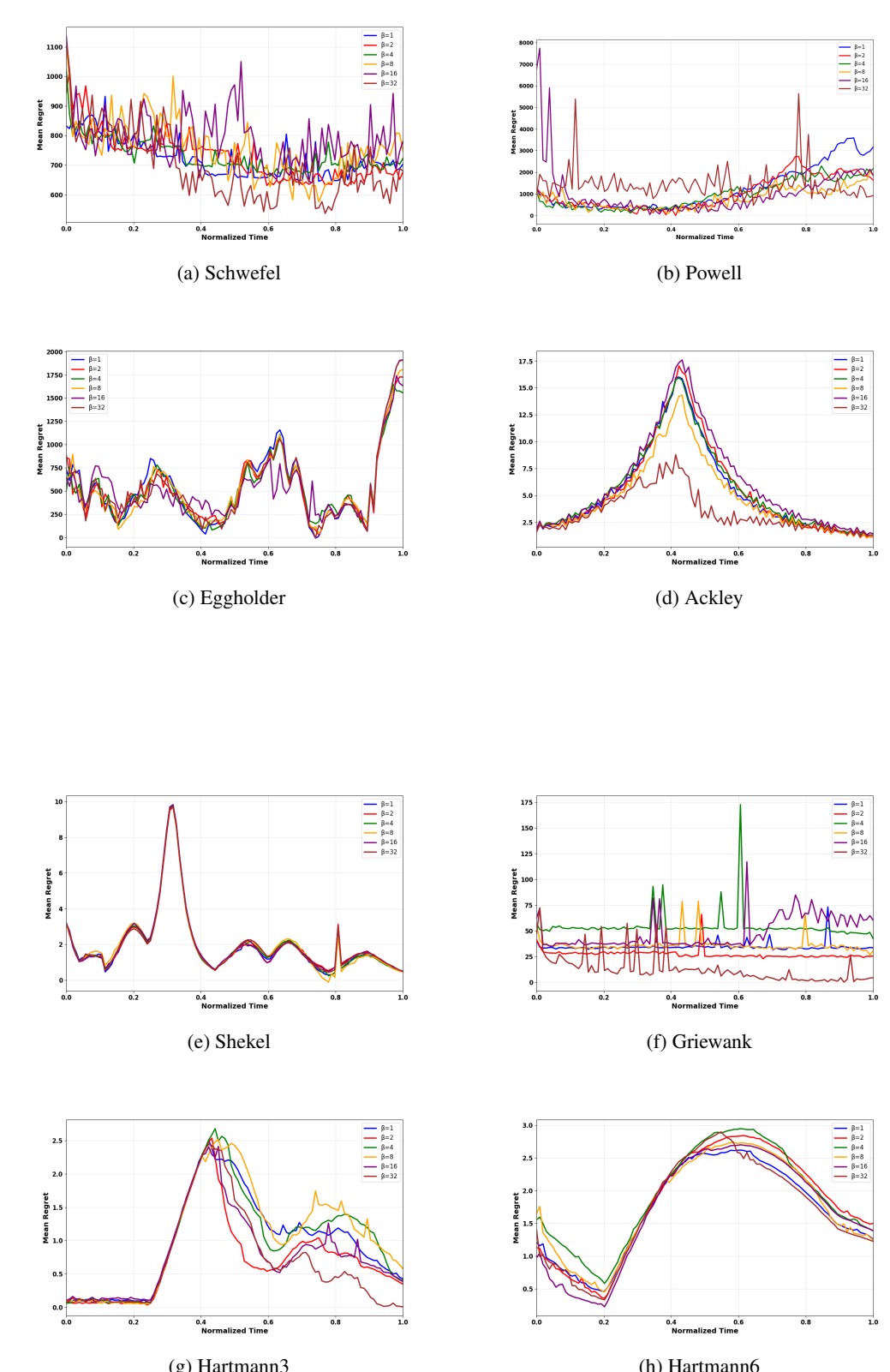

Figure 9: All eight figures show the mean regret with time for all eight simulations for the WDBO-BOBA model with a fixed observation horizon. All figures show how adapting the $\beta$ value changes the models ability to minimize regret.

A.1.4   WDBO-BOBA (N) FIXED OBSERVATIONS

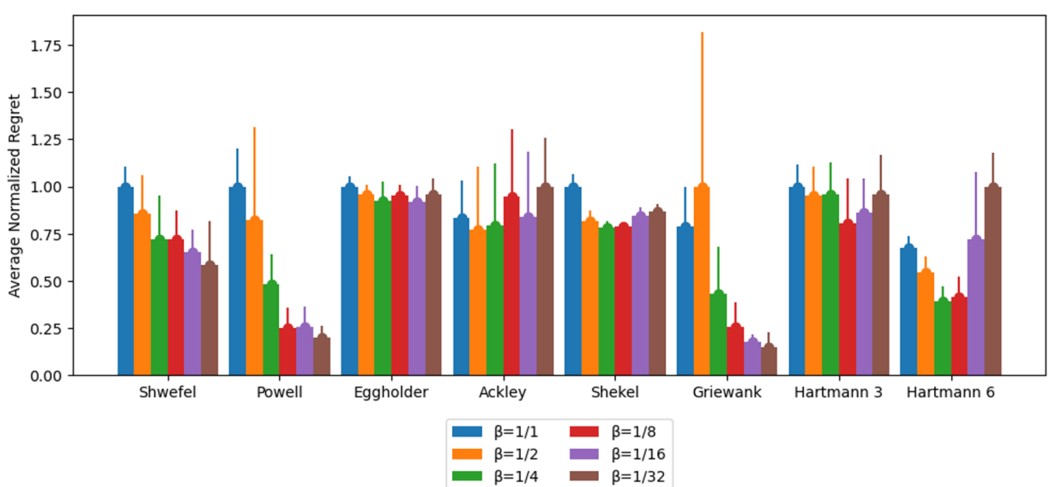

Figure 10: A grouped bar chart showing the variance $\beta$ has on the WDBO-BOBA (N) model with fixed observation horizon for all simulations

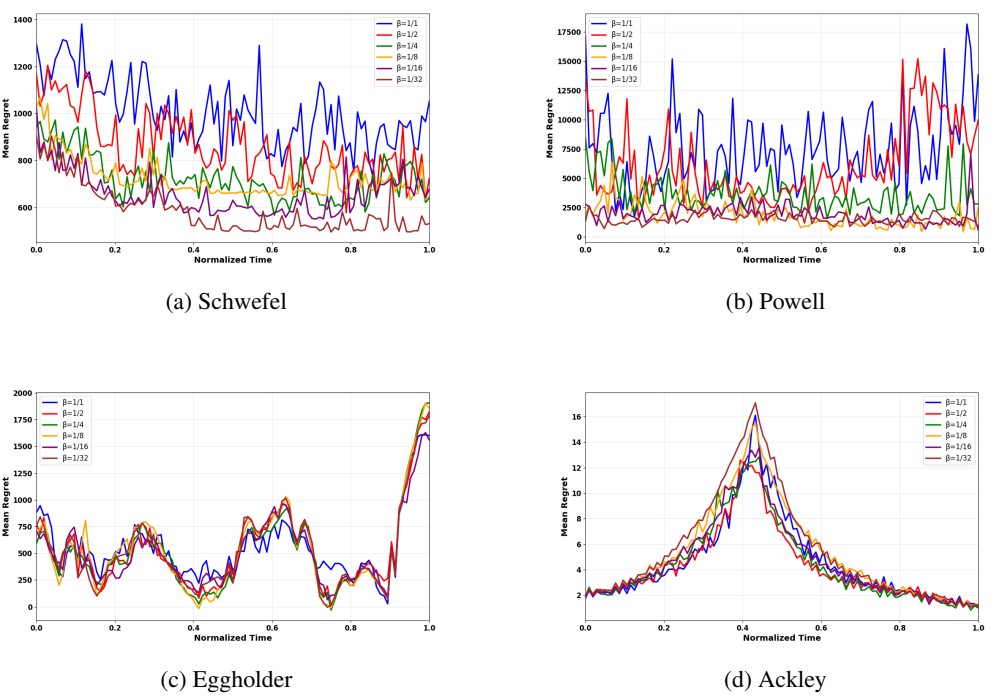

(a) Schwefel

(b) Powell

(c) Eggholder

(d) Ackley

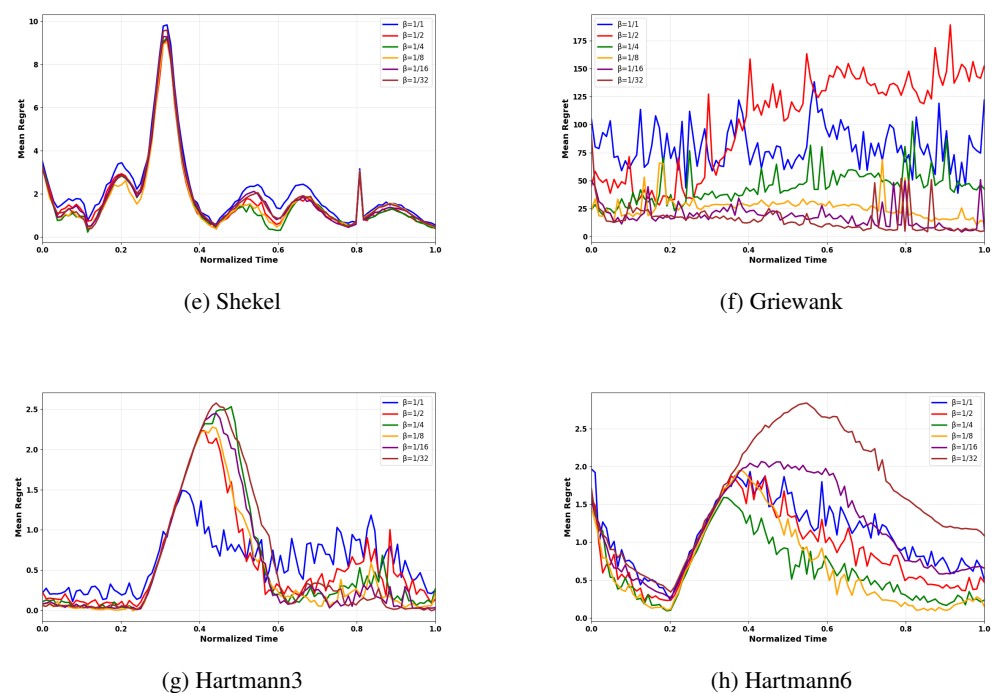

(e) Shekel

(f) Griewank

(g) Hartmann3

(h) Hartmann6

Figure 11: All eight figures show the mean regret with time for all eight simulations for the WDBO-BOBA (N) model with a fixed observation horizon. All figures show how adapting the $\beta$ value changes the models ability to minimize regret.

### A.1.5 GP-BOBA (N) FIXED TIME

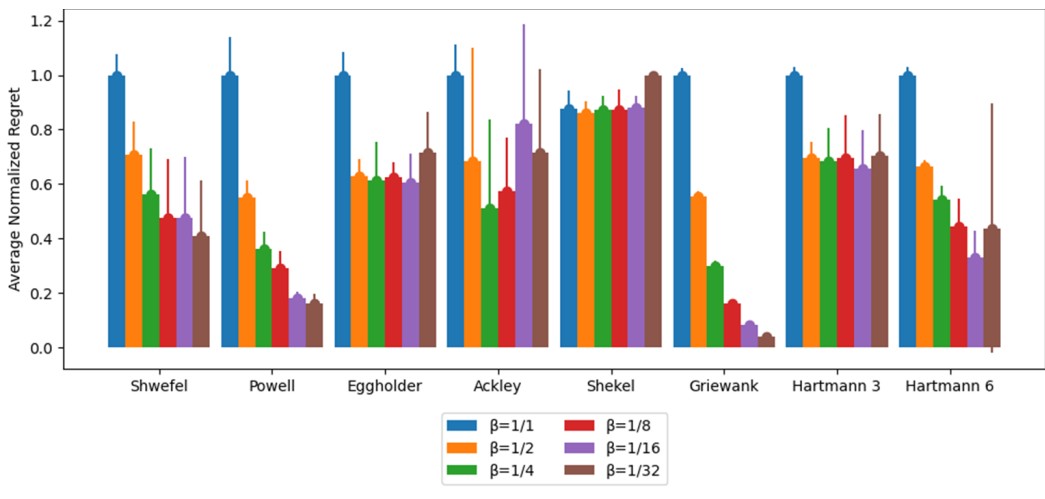

Figure 12: A grouped bar chart showing the variance $\beta$ has on the GP-BOBA (N) model with fixed time horizon for all simulations

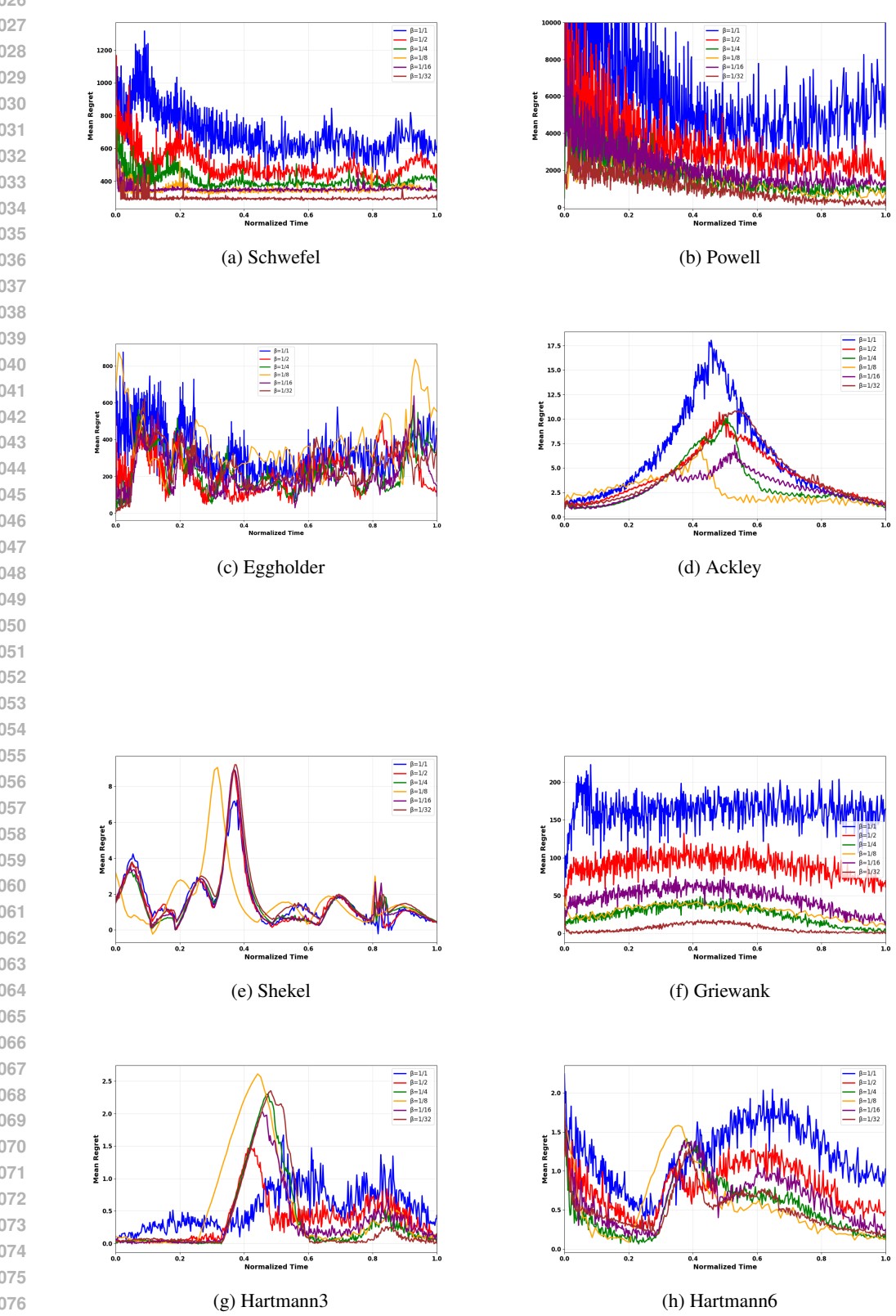

Figure 13: All eight figures show the mean regret with time for all eight simulations for the GP-BOBA (N) model with a fixed time horizon. All figures show how adapting the $\beta$ value changes the models ability to minimize regret.

### A.1.6 GP-BOBA FIXED TIME

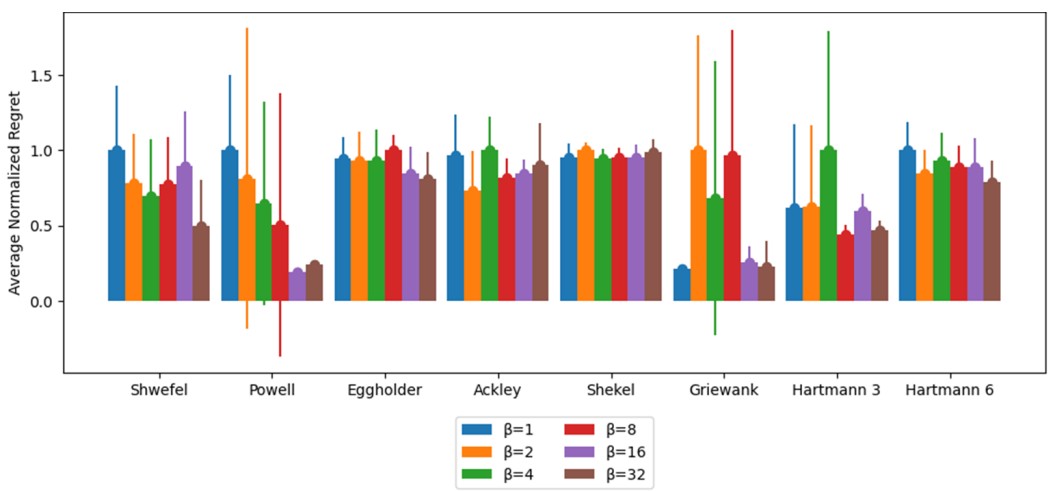

Figure 14: A grouped bar chart showing the variance $\beta$ has on the GP-BOBA model with fixed time horizon for all simulations

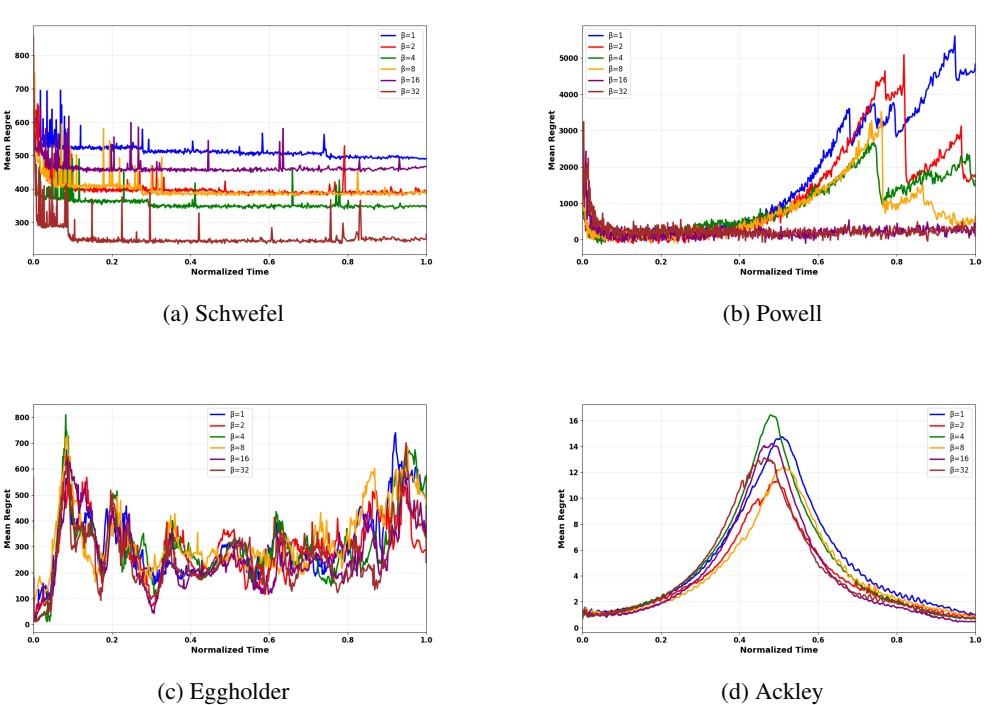

(a) Schwefel

(b) Powell

(c) Eggholder

(d) Ackley

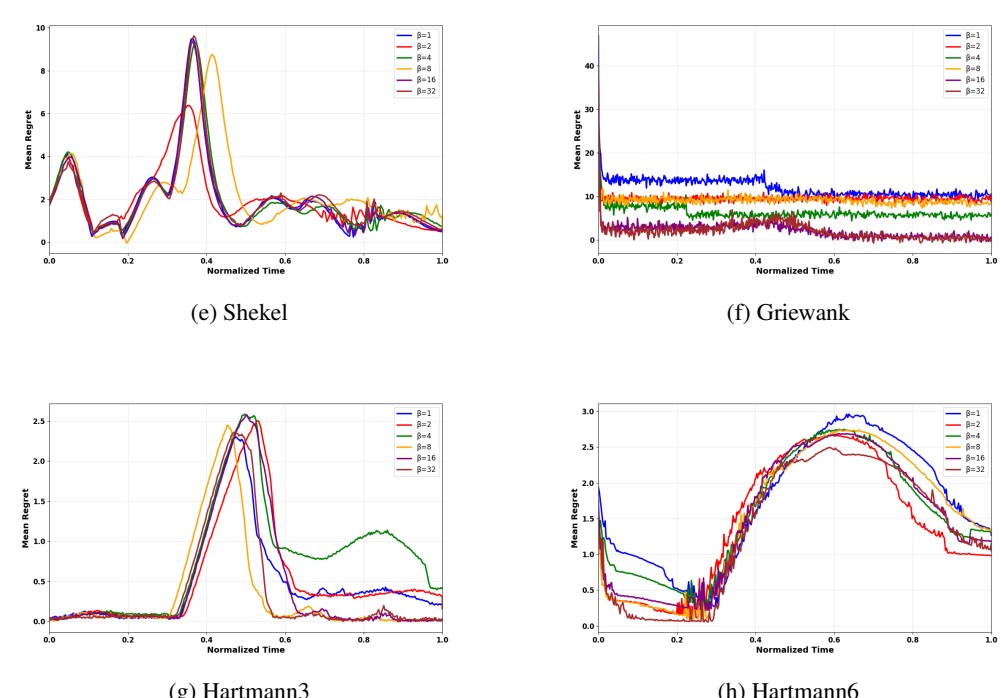

(e) Shekel

(f) Griewank

(g) Hartmann3

(h) Hartmann6

Figure 15: All eight figures show the mean regret with time for all eight simulations for the GP-BOBA model with a fixed time horizon. All figures show how adapting the $\beta$ value changes the models ability to minimize regret.

### A.1.7 WDBO-BOBA FIXED TIME

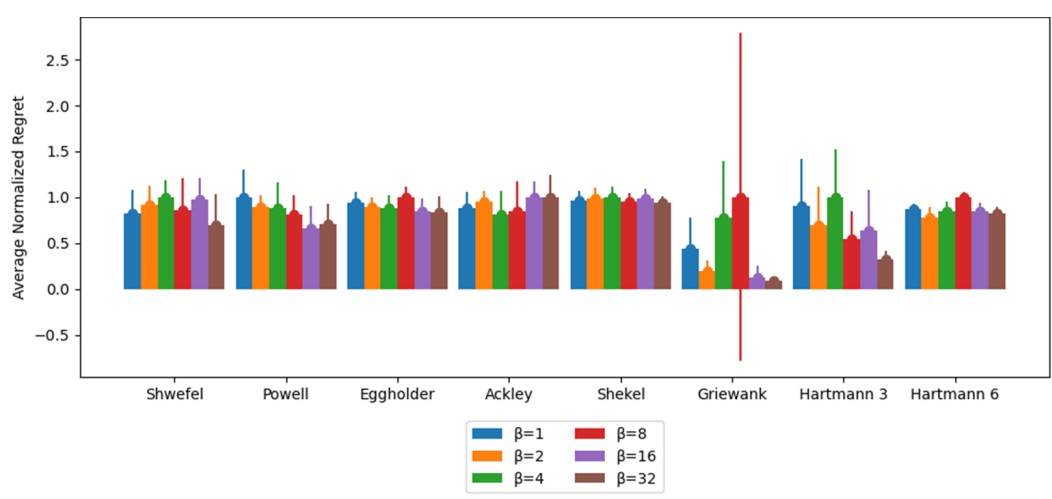

Figure 16: A grouped bar chart showing the variance $\beta$ has on the WDBO-BOBA model with fixed time horizon for all simulations

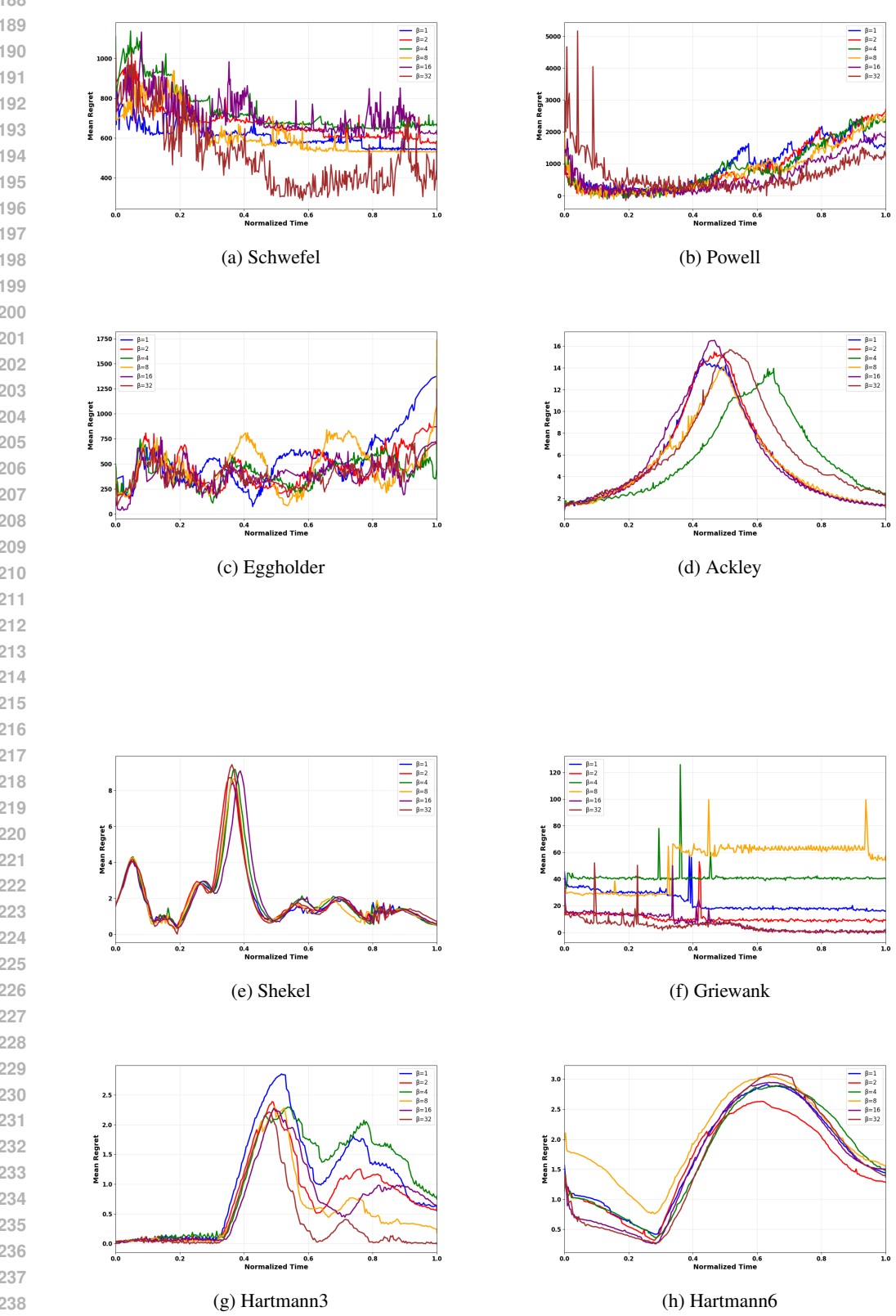

Figure 17: All eight figures show the mean regret with time for all eight simulations for the WDBO-BOBA model with a fixed time horizon. All figures show how adapting the $\beta$ value changes the models ability to minimize regret.

### A.1.8 WDBO-BOBA (N) FIXED TIME

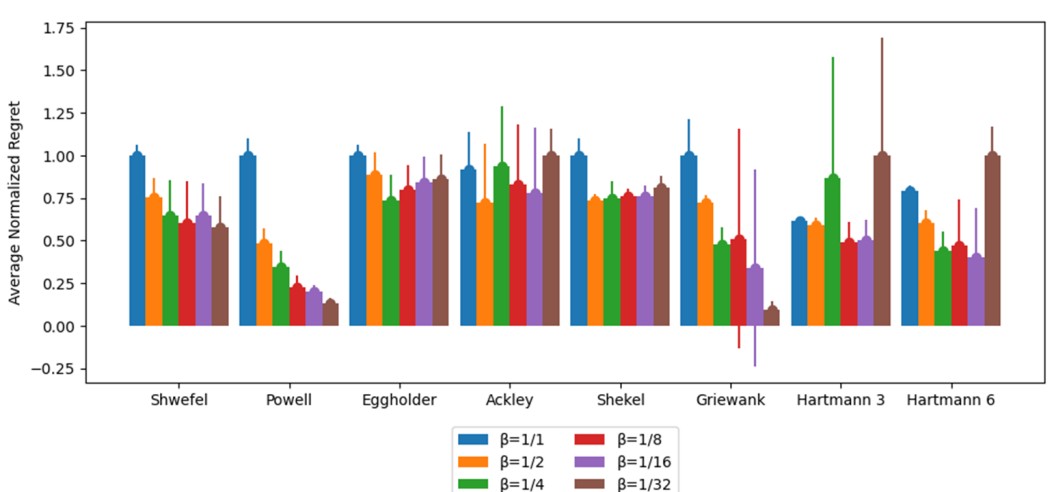

Figure 18: A grouped bar chart showing the variance $\beta$ has on the WDBO-BOBA (N) model with fixed time horizon for all simulations

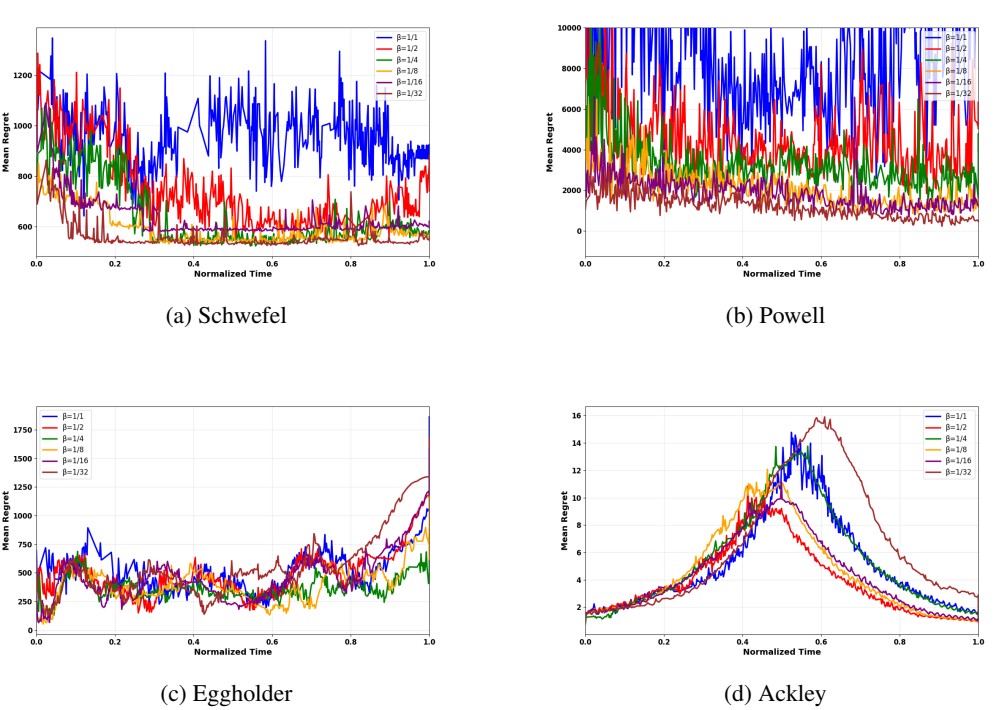

(a) Schwefel

(b) Powell

(c) Eggholder

(d) Ackley

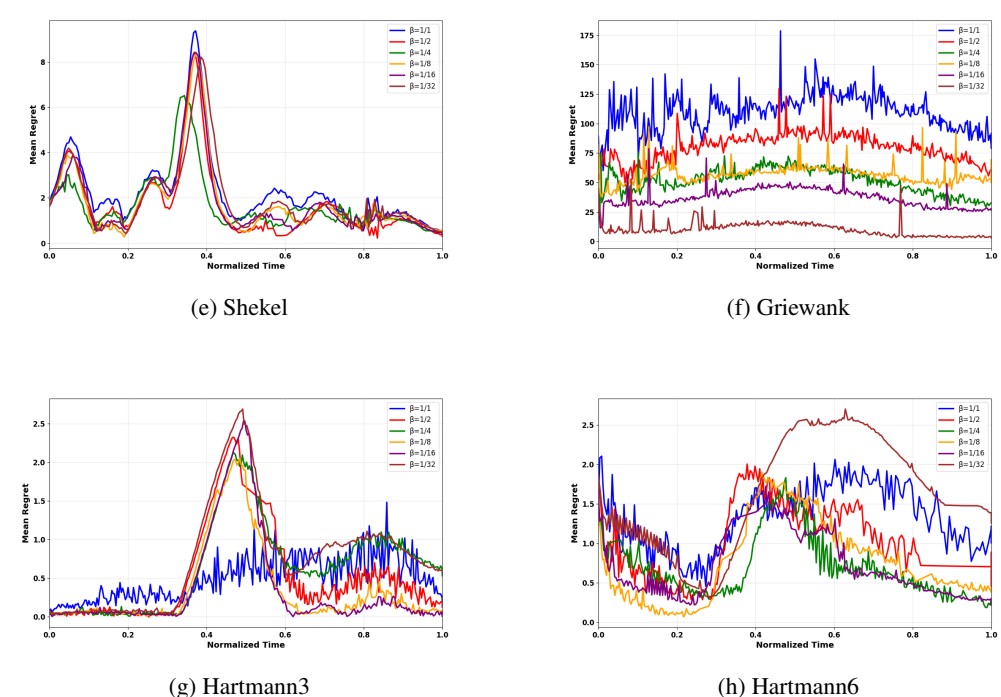

(e) Shekel  (f) Griewank

(g) Hartmann3  (h) Hartmann6

Figure 19: All eight figures show the mean regret with time for all eight simulations for the WDBO-BOBA (N) model with a fixed time horizon. All figures show how adapting the $\beta$ value changes the models ability to minimize regret.

## A.2 EMPIRICAL RESULTS

In each benchmark, dimension $d$ consists of space and time where space is the first $d-1$ dimensions and time is always the last dimension. All dimensions are normalized so $d \in [0,1]^d$ based on specific bound for each simulation. Time is normalized based on if the simulation has fixed observations or fixed time. Fixed time normalizes $t$ using min-max normalization from a range of $[0, 600]$. Fixed observation has 120 evenly spaced observations ranging from $[0, 1]$. All synthetic experiments had a normally distributed gaussian noise of 2.5% of the objective function variance.

For each model, the Matérn-5/2 kernel was used for all space dimensions. GP-UCB, GP-BOBA, and GP-BOBA(N) only had a single kernel and treated time as a spatial kernel. The Matérn-3/2 kernel was applied to the temporal dimension for WDBO, WDBO-BOBA and WDBO-BOBA(N) models.

Each BO model starts with 15 random queries which was then followed by a query that was determined by the UCB or BOBA function. All benchmark models were implemented using the BOTorch library (cite) and followed the same process to prevent differences in outcomes. All code was implemented using python except WDBO which has a backend of C++ to compute the computationally costly relevancy criteria which was then bound to python. All experiments were replicated ten times independently with an NVIDIA GeForce RTX 3090. Each model was validated 10 times and the mean regret was used to determine each models effectiveness.

Below are a list of all experiments used in this paper:

### A.2.1 SCHWEFEL (4):

$$f(\mathbf{x}) = 418.9829n - \sum_{i=1}^{n} x_i \sin\left(\sqrt{|x_i|}\right)$$

Bounds for this equation were selected by $[-500.0, 500.0]^4$ with $f(\mathbf{x}^*) = 0$ at $\mathbf{x}^* = (420.9687, \dots, 420.9687)$.

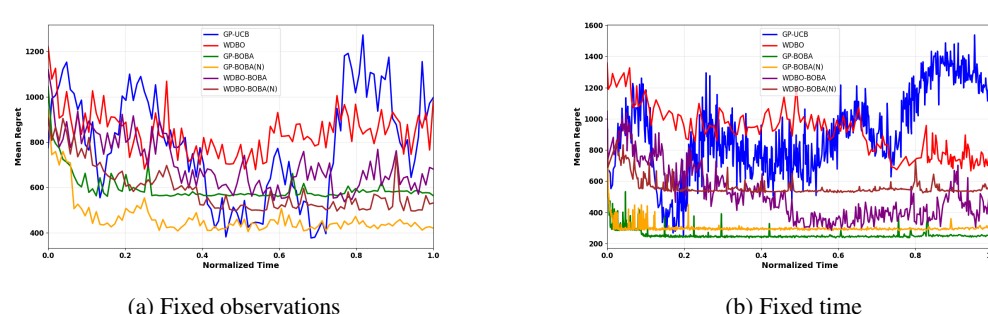

(a) Fixed observations          (b) Fixed time

Figure 20: Both figures show how different models performed based on the metric mean regret over time on the Schwefel function.

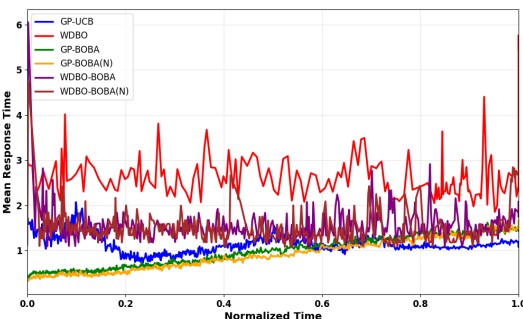

Figure 21: Demonstrates each models mean response time between observations in the fixed time horizon setting over time for the Schwefel function.

**Interpretation:** The Schwefel functions optimal point at each time step is constant at 420.9687 for each input. This would require each model to identify this point through exploration and maintain this input to minimize regret. As shown in Figure 20, only GP-BOBA and GP-BOBA (N) achieve a small continuous regret whereas other models such as GP-UCB fluctuate especially at the end of the simulations.

### A.2.2 POWELL (4):

$$f(\mathbf{x}) = (x_1 + 10x_2)^2 + 5(x_3 - x_4)^2 + (x_2 - 2x_3)^4 + 10(x_1 - x_4)^4$$

Bounds for this equation were selected by $[-4.0, 5.0]^4$ with $f(\mathbf{x}^*) = 0$ at $\mathbf{x}^* = (0, \ldots, 0)$.

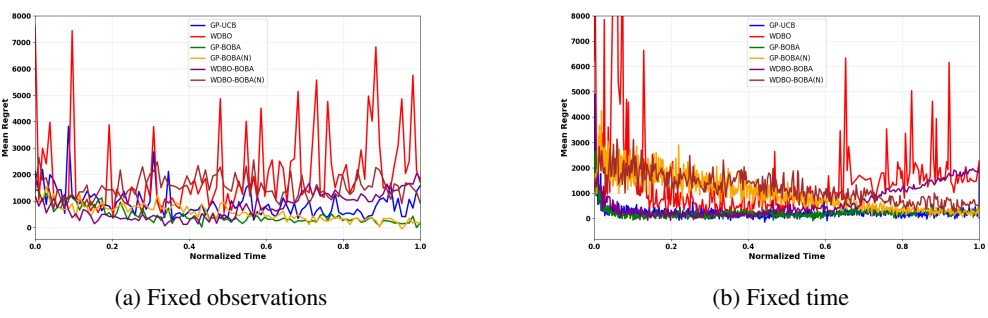

(a) Fixed observations          (b) Fixed time

Figure 22: Both figures show how different models performed based on the metric mean regret over time on the Powell function.

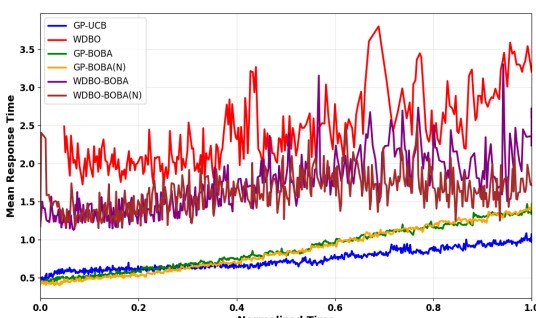

Figure 23: Demonstrates each models mean response time between observations in the fixed time horizon setting over time for the Powell function.

**Interpretation:** The Powell functions optimal point requires a balance between all inputs especially the first dimension and time. This function requires small changes to maintain minimum regret whereas large changes from exploration would lead to larger regret accumulation. As shown in Figure 22, when observations were fixed, both GP-BOBA and GP-BOBA (N) identify the optima and maintain a low regret bound whereas models such as WDBO which explores due to observation removal has large increases in regret. Similar findings are repeated in the fixed time setting, especially for WDBO which increases regret through time. However, the GP-UCB performance improves due to its ability to make more observations in the allotted time to match the performance of GP-BOBA.

### A.2.3   EGGHOLDER (2):

$$f(x) = -(x_2 + 47)\sin\sqrt{\left|x_2 + \frac{x_1}{2} + 47\right|} - x_1 \sin\sqrt{|x_1 - (x_2 + 47)|}$$

Bounds for this equation were selected by $[-512.0, 512.0]^2$ with $f(\mathbf{x}^*) = 959.6407$ at $\mathbf{x}^* = (512, 404.2319)$.

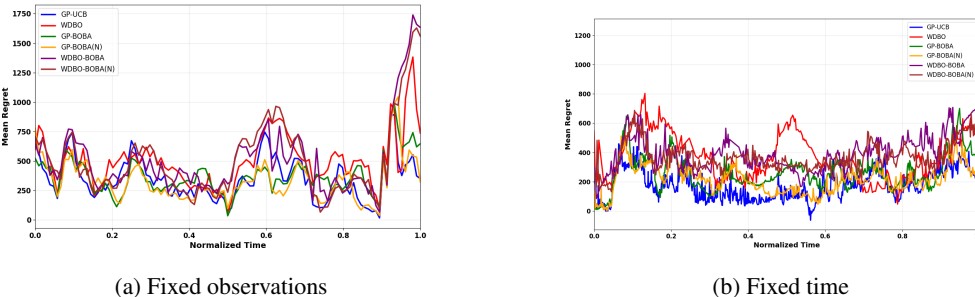

(a) Fixed observations                    (b) Fixed time

Figure 24: Both figures show how different models performed based on the metric mean regret over time on the Eggholder function.

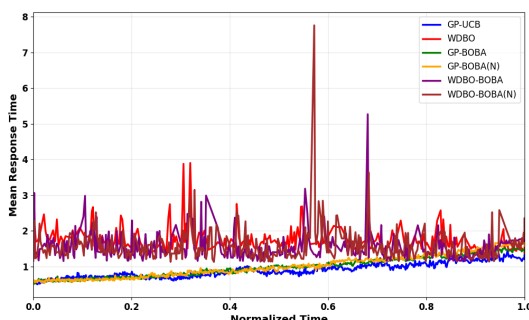

Figure 25: Demonstrates each models mean response time between observations in the fixed time horizon setting over time for the Eggholder function.

**Interpretation:** The Eggholder function has many local optima which require exploration to minimize regret. In the fixed observations horizon, where all models appear to follow a similar pattern except at the end of the function where both GP-UCB and GP-BOBA (N) are able to identify the global optima closer to the other models. The relationship changes significantly in the fixed time horizon where GP-UCB is able to minimize regret significantly compared to the other models. GP-BOBA (N) has similar performance but accumalates significantly more regret compared to GP-UCB.

A.2.4 ACKLEY (4):

$$f(\mathbf{x}) = -20 \exp\left(-0.2\sqrt{\frac{1}{d}\sum_{i=1}^{d} x_i^2}\right) - \exp\left(\frac{1}{d}\sum_{i=1}^{d}\cos(2\pi x_i)\right) + 20 + e$$

Bounds for this equation were selected by $[-32.0, 32.0]^4$ with $f(\mathbf{x}^*) = 0$ at $\mathbf{x}^* = (0, \ldots, 0)$.

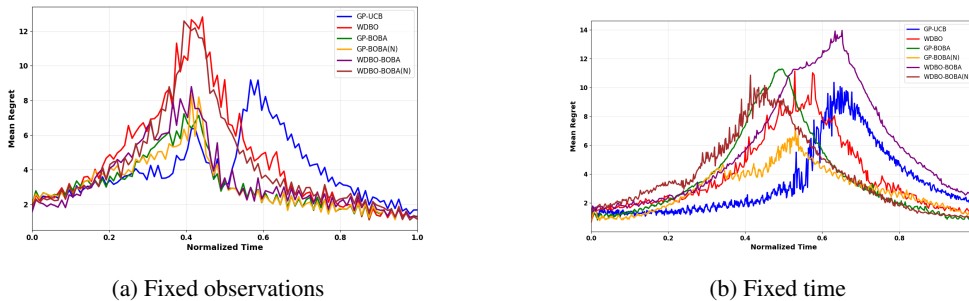

(a) Fixed observations         (b) Fixed time

Figure 26: Both figures show how different models performed based on the metric mean regret over time on the Ackley function.

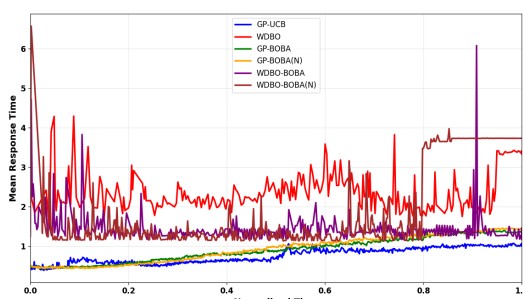

Figure 27: Demonstrates each models mean response time between observations in the fixed time horizon setting over time for the Ackley function.

**Interpretation:** The Ackley function has a large global optima in the middle of the function that occurs at 0.5 normalized time. Many functions increase regret once normalized time approaches 0.5 but most functions except WDBO and WDBO-BOBA (N) identify the global optima and in turn decrease their regret. However, GP-UCB once identified the optima increases regret suddenly, impacting performance whereas all other models find a similar policy which is close to the local optima for the following time steps. This relationship can be seen for the fixed observation and time horizon where GP-BOBA (N) identifies the global and local optima over time.

A.2.5   SHEKEL (4):

$$f(\mathbf{x}) = -\sum_{i=1}^{10} \frac{1}{\sum_{j=1}^{4}(x_j - C_{ij})^2 + \beta_i}$$

where $\beta = \frac{1}{10}[1, 2, 2, 4, 4, 6, 3, 7, 5, 5]^T$ and

$$\mathbf{C} = \begin{pmatrix} 4.0 & 1.0 & 8.0 & 6.0 & 3.0 & 2.0 & 5.0 & 8.0 & 6.0 & 7.0 \\ 4.0 & 1.0 & 8.0 & 6.0 & 7.0 & 9.0 & 3.0 & 1.0 & 2.0 & 3.6 \\ 4.0 & 1.0 & 8.0 & 6.0 & 3.0 & 2.0 & 5.0 & 8.0 & 6.0 & 7.0 \\ 4.0 & 1.0 & 8.0 & 6.0 & 7.0 & 9.0 & 3.0 & 1.0 & 2.0 & 3.6 \end{pmatrix}$$

Bounds for this equation were selected by $[0.0, 10.0]^4$ with $f(\mathbf{x}^*) = 10.5364$ at $\mathbf{x}^* = (4, \ldots, 4)$.

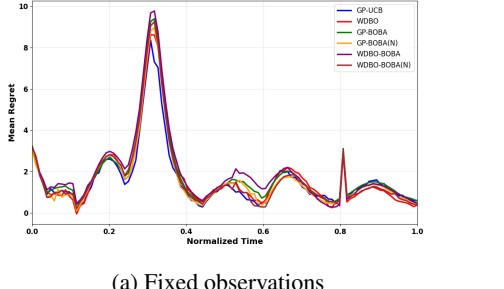

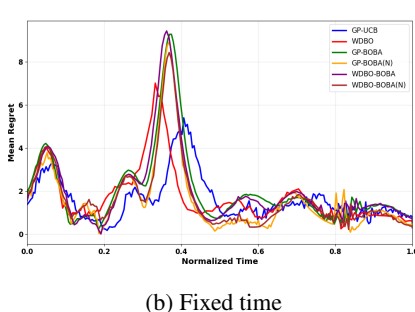

(a) Fixed observations                    (b) Fixed time

Figure 28: Both figures show how different models performed based on the metric mean regret over time on the Shekel function.

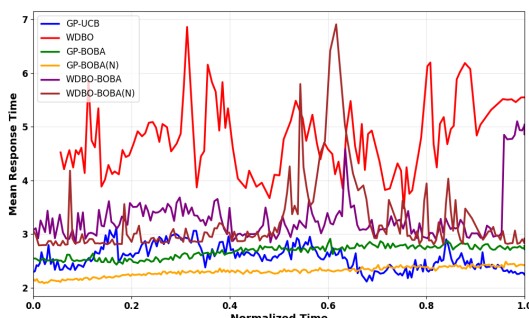

Figure 29: Demonstrates each models mean response time between observations in the fixed time horizon setting over time for the Shekel function.

**Interpretation:** The Shekel function is a sparse function that has peaks with different coordinates at different time points that must be identified by exploration. Due to the sparsity of this function, the fixed observation horizon performance for all models are extremely similar, showing that this function is difficult to optimize in this setting regardless of the model chosen. However, this is not the case for the fixed time horizon where GP-UCB and GP-BOBA (N) outperform all other models. What distinguishes these high performing models to the others is GP-UCB identifies the global optima at normalized time 0.4 quicker than all models and GP-BOBA (N) maintains an overall lower regret bound across the function.

### A.2.6 GRIEWANK (6):

$$f(\mathbf{x}) = 1 + \frac{1}{4000} \sum_{i=1}^{6} x_i^2 - \prod_{i=1}^{6} \cos\left(\frac{x_i}{\sqrt{i}}\right)$$

Bounds for this equation were selected by $[-600.0, 600.0]^6$ with $f(\mathbf{x}^*) = 0$ at $\mathbf{x}^* = (0, \ldots, 0)$.

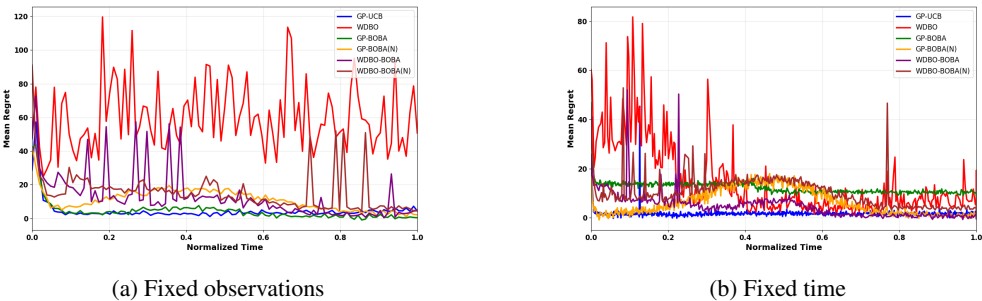

(a) Fixed observations          (b) Fixed time

Figure 30: Both figures show how different models performed based on the metric mean regret over time on the Griewank function.

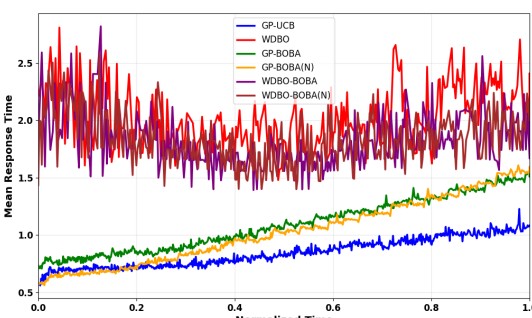

Figure 31: Demonstrates each models mean response time between observations in the fixed time horizon setting over time for the Griewank function.

**Interpretation:** The Griewank function represents a bowl shape with many local minima across the surface which has an optima in the center of the function which does not require extensive exploration. In the fixed observation horizon, both GP-UCB, GP-BOBA and GP-BOBA (N) identify the local optima point faster than the WDBO based models. However, only GP-UCB and GP-BOBA maintain this level of regret whereas GP-BOBA (N) continues exploring, leading to a temporary increase in regret. This relationship is mimicked in the fixed time horizon where GP-BOBA (N)'s performance deteriorates in the middle of the function. Unexpectedly, even though GP-UCB's performance is improved with more observations, GP-BOBA does not as it appears to maintain regret throughout the function. This may be due to the high dimensionality of this function but further explanations would be required to understand why.

### A.2.7 HARTMANN3 (3):

$$f(\mathbf{x}) = -\sum_{i=1}^{4} \alpha_i \exp\left(-\sum_{j=1}^{3} A_{ij}(x_j - P_{ij})^2\right)$$

where $\alpha = [1.0, 1.2, 3.0, 3.2]$ and

$$A = \begin{pmatrix} 3 & 10 & 30 \\ 0.1 & 10 & 35 \\ 3 & 10 & 30 \\ 0.1 & 10 & 35 \end{pmatrix}, \quad P = 10^{-4} \begin{pmatrix} 3689 & 1170 & 2673 \\ 4699 & 4387 & 7470 \\ 1091 & 8732 & 5547 \\ 381 & 5743 & 8828 \end{pmatrix}$$

Bounds for this equation were selected by $[0.0, 1.0]^3$ with $f(\mathbf{x}^*) = 3.86278$ at $\mathbf{x}^* = (0.114614, 0.555649, 0.852547)$.

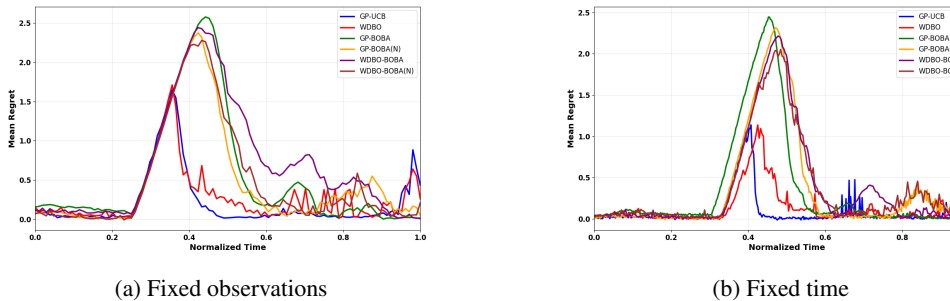

(a) Fixed observations        (b) Fixed time

Figure 32: Both figures show how different models performed based on the metric mean regret over time on the Hartmann 3 function.

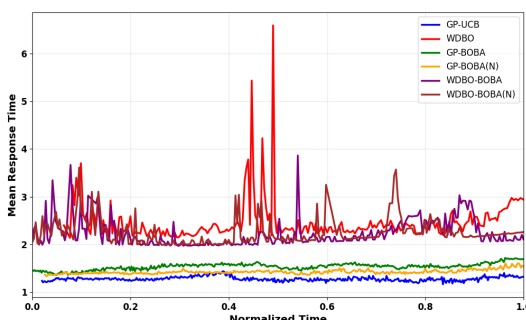

Figure 33: Demonstrates each models mean response time between observations in the fixed time horizon setting over time for the Hartmann3 function.

**Interpretation:** The Hartmann3 function is an exponential function with 4 local optima. The local optima at time 0.3 - 0.5 normalized time which is the reason why all BOBA models perform poorly. Shown from the figures in section A.1 that shows how $\beta$ affects performance for Hartmann3, it can be seen that $\beta$ values that increase exploration does not encounter high regret during this local optima. However, these models with greater exploration performance deteriorates after this region whereas the BOBA models compared to GP-UCB does not, showing the limitations of having a fixed hyperparameter value for functions similar to the Hartmann3 independent of the fixed horizon chosen.

### A.2.8 HARTMANN6 (6):

$$f(\mathbf{x}) = -\sum_{i=1}^{4} \alpha_i \exp\left(-\sum_{j=1}^{6} A_{ij}(x_j - P_{ij})^2\right)$$

where $\alpha = [1.0, 1.2, 3.0, 3.2]$ and

$$A = \begin{pmatrix} 10 & 3 & 17 & 3.50 & 1.7 & 8 \\ 0.05 & 10 & 17 & 0.1 & 8 & 14 \\ 3 & 3.5 & 1.7 & 10 & 17 & 8 \\ 17 & 8 & 0.05 & 10 & 0.1 & 14 \end{pmatrix}, \quad P = 10^{-4} \begin{pmatrix} 1312 & 1696 & 5569 & 124 & 8283 & 5886 \\ 2329 & 4135 & 8307 & 3736 & 1004 & 9991 \\ 2348 & 1451 & 3522 & 2883 & 3047 & 6650 \\ 4047 & 8828 & 8732 & 5743 & 1091 & 381 \end{pmatrix}$$

Bounds for this equation were selected by $[0.0, 1.0]^6$ with $f(\mathbf{x}^*) = 3.32237$ at $\mathbf{x}^* = (0.20169, 0.150011, 0.476874, 0.275332, 0.311652, 0.6573)$.

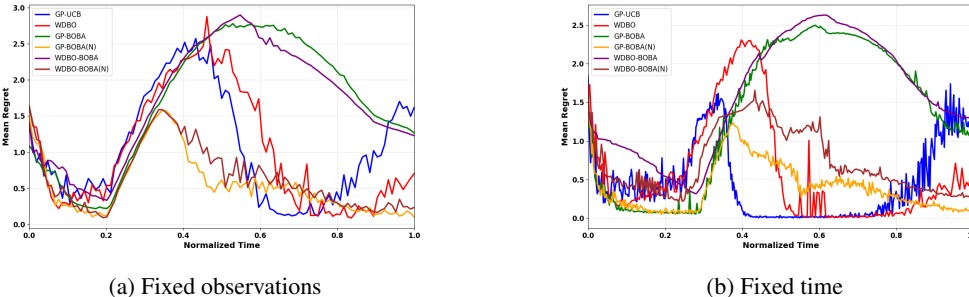

(a) Fixed observations          (b) Fixed time

Figure 34: Both figures show how different models performed based on the metric mean regret over time on the Hartmann 6 function.

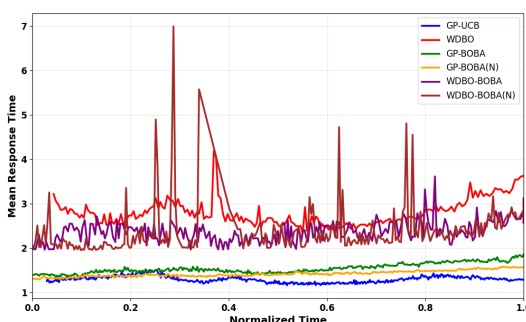

Figure 35: Demonstrates each models mean response time between observations in the fixed time horizon setting over time for the Hartmann6 function.

**Interpretation:** The Hartmann6 function is similar to the Hartmann3, except it has 3 more dimensions and 6 local optima. However, unlike Hartmann3 where BOBA consistently accumulates more regret, this is not the case for this function. In the fixed observation horizon, both GP-BOBA (N) and WDBO-BOBA (N) both are able to explore to identify the local optima at normalized time 0.4 whereas all other models take a longer period of time to reduce their mean regret. In the fixed time horizon, GP-UCB's performance improves during this range and identifies the local optima between the normalized time 0.4 - 0.8 which is not replicated by other models. Nonetheless, GP-UCB's regret increases between 0.8 - 1.0 normalized time due to a lack of identification of the final optima whereas GP-BOBA (N) which accumulates less regret maintains a steady decrease of mean regret from 0.4 - 1.0.

### A.3 MODEL PERFORMANCE

#### A.3.1 GP-UCB

GP-UCB was chosen as many other papers use this model as a benchmark to compare performance to the state of the art (Bardou & Thiran, 2025; Nyikosa et al., 2018). For dynamic environments, GP-UCB treats time as a space dimension where GP-UCB can only observe queries in the current time as opposed to the past and the future.

It was unexpected that GP-UCB would perform to this level in this paper based on previous findings (Bardou & Thiran, 2025; Bardou et al., 2024). As shown in Figure 23 and similar figures that show mean response time of each model, GP-UCB remains smaller with respect to other models even without an observation removal policy. The authors hypothesise that this smaller response time is the reason why GP-UCB performs exceptionally well on these simulations.

#### A.3.2 WDBO

WDBO was selected from Bardou et al. (2024) due to its state of the art performance against benchmarked models in tests similar to the simulations chosen in this paper. However, performance in Bardou et al. (2024) paper does not match the findings of our paper. We believe this is due to the fact that Bardou et al. (2024) used their CPU on BOTorch whereas we use a GPU during evaluation. From their results, both the GP-UCB and WDBO models take approximately the same response time of 2 seconds. In this paper this is not the case as the time it takes WDBO to make one observation, on average the GP-UCB model can make four observations.

From these findings, we assume the GP-UCB as coded completely with BOTorch can take full advantage of the GPUs used in this paper whereas WDBO's performance does not appear to be affected by the GPU based on the response time reported by Bardou et al. (2024). However, this does not negate WDBO's previous performance as Bardou & Thiran (2025) demonstrated response time is a significant factor on regret. Additionally, different applications for DBO will require either CPU or GPU depending on access and the black box the the DBO is attempting to optimize.

### A.3.3 GP-BOBA

Both GP-BOBA and GP-BOBA (N) performed significantly better than the other models especially in the fixed observation horizon settings, demonstrating their aptitude in restricted settings in comparison with standard models. In the fixed observation horizon, the Hartmann3 test is the only function where other models outperform all BOBA models. This is due to the limitation of a fixed $\beta$ value as shown in section A.1.

In fixed time horizon settings, GP-BOBA has the best performance in half of all functions showing that the BOBA acquisition function is also suitable for non-restricted settings. The speculated contributing reasons why GP-UCB outperforms GP-BOBA in these settings is due to the smaller mean response time in GP-UCB and the risk of overfitting the GP will negatively affect GP-BOBA's performance. In future works, different methods to mitigate these reasons may be used to see if BOBA's performance increases compared to GP-UCB. Additionally, by removing the requirement of the $\beta$ parameter so that the BOBA functions can increase exploration or exploitation automatically depending on the function and its time step, it is assumed BOBA's performance will increase for all function and both fixed horizon settings.

### A.3.4 WDBO-BOBA

WDBO-BOBA and WDBO-BOBA (N) has a similar relationship GP-BOBA has with GP-UCB where the BOBA function improves model performance especially in the fixed observation horizon setting where WDBO-BOBA has results with non-significant difference to the best performing models. In the fixed time horizon, the BOBA function significantly enhanced the WDBO model for half of the functions but not the same function as GP-BOBA did.

As stated earlier, we speculate the reason why WDBO does not perform as expected from previous benchmarks due to computational differences (Bardou et al., 2024). However, we can assume that as WDBO-BOBA enhanced the performance of the WDBO model, that these results will also be replicated with different hardware.

