# OpenReview forum: "BOBA: Dynamic Bayesian Optimization through Bayesian Active Inference"
_ICLR.cc/2026/Conference — Submitted to ICLR 2026_

### Official Review · Reviewer_daMT · 2025-10-21

**Soundness:** 1
**Presentation:** 1
**Contribution:** 2
**Rating:** 2
**Confidence:** 4

**Summary:**

The paper proposes a novel acquisition function for time-varying Bayesian optimisation, inspired by the free energy principle.  The authors derive the acquisition function and claim it should be particularly suitable for dynamic environments, where the time intervals are not evenly spaced. They then proceed to evaluate their algorithm on synthetic problems.

**Strengths:**

- The idea of combining Active Inference in DBO is, to my knowledge, novel and interesting

**Weaknesses:**

- The authors admit that the algorithm's performance depends heavily on the hyperparameter $\beta$ and say they tuned it for each problem separately. However, BO problem setting is a zero-shot one, that is, we want to optimise an unknown black-box function with a limited budget and we do not get any "validation" set on we can fine-tune parameters. As such, all methods have to work straight out of the box, which is the main difficulty of this problem. As such, the need for accurate tuning of this parameter make this method effectively useless in practice
- The algorithm also requires to know the prior over the optimum $P(\tilde{o})$, but if the function is truly black-box, how can we have any information about the location of the optimum? Again, this casts a shadow on practical usability of the algorithm.
- The proposed algorithm seems to be underperforming to GP-UCB on the experiments, especial in Table 2, so in the setting where time intervals vary. This is the setting that was advertised at the beginning of the paper as the one were proposed method should particularly excel, so the fact the simplest, off-shelf GP-UCB outperforms your method unfortunately puts in question the relevance of your idea.

**Questions:**

- In Assumption 1, what is meant by $GP_t \in \mathbb{R}^n$? A Gaussian Process is a stochastic process, that is essentially a measure over a space of function. How can a measure be a vector? This notation is confusing and not clear enough.
- In Assumption 2, you then write $GP_t \in \mathcal{N}(0, 1)$, a distribution is not a set, how can something belong to a distribution? Did you mean to write $\sim$ instead of $\in$?
- In the main body you say that the way in which you choose $P(\tilde{o})$ is explained in the Appendix. However, I couldn't find it anywhere. Could you please point me to where it is (page number and line numbers)?
- You claim to have shared your code, but the link seems malformed. You provided the link  https/GitHub/Anonymous, which is not even a valid url. Could you please share your code?

---

### Official Review · Reviewer_377e · 2025-10-23

**Soundness:** 1
**Presentation:** 1
**Contribution:** 2
**Rating:** 2
**Confidence:** 3

**Summary:**

This paper proposes a new acquisition function called "Bayesian Optimization through Bayesian Active inference" (BOBA) for time-varying Bayesian Optimization (TVBO), also known as Dynamic Bayesian Optimization (DBO).

The acquisition function is derived from active inference and free energy principles and it is claimed to enable a better querying policy for the BO algorithm by achieving better exploration-exploitation trade-offs in dynamic environments.

BOBA is benchmarked on time-varying synthetic problems against some TVBO algorithms.

**Strengths:**

* Finding more informative acquisition functions is of strong interest to the BO community, and this is particularly true in time-varying settings.

* Using free-energy and active inference principles to design an acquisition function is an interesting idea.

**Weaknesses:**

Although the ideas in this paper are interesting, I do not think the paper meets the quality standard of a top-tier conference like ICLR. I detail my point below, with labelled weaknesses to facilitate the authors' response during the discussion period.

1. **Writing quality.** I think the paper's writing quality could be massively improved. A lot of formulations are vague, this prevents me to understand the paper and its core contribution. Here is a (non exhaustive) list of examples:
	* Assumption 1 feels very arbitrary, and this is particularly the case of the recommended $n = 262144^{1/(d-1)}$. The recommendation is justified directly in the Assumption statement, this makes it difficult to distinguish what is assumed and why it is assumed. Also, I believe that Assumption 1 is very restrictive and likely does not hold in practice.

	* I cannot make sense of Assumption 2.

	* Some mathematical expressions are nonstandard and/or do not make sense, e.g., the operator $\arg\max$ is written $argmax$, $\mathbf{GP_t}$ seems to be an element of $\mathbb{R}^n$ (Assumption 1) but is later written as an "element" of the standard normal distribution $\mathcal{N}(0, 1)$ (Assumption 2), which is used as a set...

2. **Insufficiently supported claims.** In the abstract and the conclusion, the paper claims that AIF and BOBA clearly improve the performance of TVBO. However, looking at Tables 1 and 2, it seems that GP-BOBA achieves performance that is roughly on par with GP-UCB. This is particularly clear in Table 2. Additionally, a theoretical study of this claim would have strengthened the paper.

3. **Experimental setting.** GP-UCB, as described in Appendix A.2, seems to to be aware of the time-varying nature of the objective function. In fact, its Matérn-5/2 kernel seems to have access to the problem parameters *and* to the temporal coordinate. If that is indeed the case, the implemented GP-UCB is not the vanilla GP-UCB used in static BO (which has only access to the problem parameters) and cannot serve as a control solution. Also, the experimental results would have been strengthened by considering other state-of-the-art TVBO algorithms (TV-GP-UCB [1], ET-GP-UCB [2]).

4. **Negative Regret?** In Figure 1(a), the mean regret for $\beta = 16$ and $\beta = 32$ seem to become negative from time to time. This is unexpected and contradicts the standard definition of regret $r_n = f(x_n, t_n) - f(x^\*\_n, t_n)$, which is clearly nonnegative when $x_n^* = \arg\min_{x \in S} f(x, t_n)$.

**References**

[1] Bogunovic, I., Scarlett, J., & Cevher, V. (2016, May). Time-varying Gaussian process bandit optimization. In Artificial Intelligence and Statistics (pp. 314-323). PMLR.

[2] Brunzema, P., von Rohr, A., Solowjow, F., & Trimpe, S. (2022). Event-triggered time-varying Bayesian optimization. arXiv preprint arXiv:2208.10790.

**Questions:**

Here are some questions to spark the discussion with the authors.

1. Can you discuss more about the reasons why BOBA is claimed to significantly improve the performance of TVBO algorithms?

2. Have you tried to formally study the cumulative regret of TVBO algorithms that use BOBA?

3. Can you confirm that GP-UCB has access to the time coordinate of each observation? If so, do you agree that it is not a control solution because it differs from the vanilla GP-UCB used in static BO?

4. Why is the regret negative in Figure 1(a)?

---

### Official Review · Reviewer_KRdK · 2025-10-29

**Soundness:** 2
**Presentation:** 1
**Contribution:** 2
**Rating:** 2
**Confidence:** 4

**Summary:**

The paper presents a new acquisition function for BO with dynamic (time-varying) objectives. The acquisition function applies ideas from active inference to the dynamic BO setting. The acquisition function has one important hyperparameter $\beta$. The method is evaluated on synthetic functions and compared to two baselines: GP-UCB and WDBO. The proposed method outperforms the baselines on average if $\beta$ is tuned accordingly.

**Strengths:**

- I believe that acquisition functions for time-varying (dynamic) BO is an interesting and relevant research topic. I also think that the information theoretic approach taken with active inference is an interesting research direction.

- I think it is good practice that the authors make explicit that their method has an important hyperparameter and examine in detail how its optimal value changes with different functions.

**Weaknesses:**

- In the results section the $\beta$ for GP-BOBA is adjusted tuned to each of the problems. However, this does not seem be the case for UCB. In fact the paper does not seem to report the value of $\beta$ chosen for GP-UCB. In my opinion it is necessary to either show GP-BOBA results with a constant $\beta$ and/or use the same method to adjust $\beta$ in UCB as in BOBA. As is, I believe it is hard to compare the performance of GP-BOBA und GP-UCB.

- For me it was hard to understand some central aspects of the work, for example:
    - Why is a softmax chosen in (15)?
    - How is the preferred observation (o \tilde) in (14) chosen? Usually one does not know the optimum value of the objective function. How would this value be chosen in practice?
    - For better understandability I would encourage the authors to include a figure that explains the main idea of the method .
    - Which aspect of the acquisition function is specifically tailored to non stationary objectives?
    - Since WDBO is one of two baseline I think it would have been nice to explain its core idea in one or two sentences in the main paper.
    - The first mention of WDBO-BOBA is in the results section. I would have expected a short explanation on how this combination works or at least a reference to the appendix.
    - Does Q() refer to a probability density function? Sometimes P() is also used to denote a PDF.

- In my opinion entropy search should have been mentioned as key related work to this paper (see first question below).

**Questions:**

- The entropy search framework, e.g., [1], has an information theoretic view on BO. It chooses the next query to minimize the entropy of the distribution of, e.g., the location of the optimum. Entropy search resolves the exploration-exploitation trade-off through the information theoretic framework itself and thus does not require an additional constant $\beta$. To me it seems that BOBA could be related to this approach. I would suggest differentiating your work from entropy search in the related work section.

- I would suggest resolving weakness 1 by adding an additional comparison to UCB.

- I would suggest clarifying the points raised in weakness 2 to improve the overall understandability of the paper.

- Problem Formulation: Equation (1) seems to be the formulation for a standard (static) black box problem. Below it is stated that we have a time-varying optimization problem. I would suggest stating the time-varying problem statement already in (1).

- How would you select $\beta$ when presented with an unknown problem which is usually the case in black-box optimization?

- Minor: There is a placeholder for the BOTorch citation in appendix A2: 'BOTorch library (cite)'
- Minor: In my opinion functions such as argmax or softmax are usually written non-italic (15)
- Minor: The text in the figures is very small and thus hard to read.

[1] Hernández-Lobato, José M., Matthew W. Hoffman, and Zoubin Ghahramani. "Predictive entropy search for efficient global optimization of black-box functions." _Advances in neural information processing systems_ 27 (2014).

---

### Official Review · Reviewer_KGfR · 2025-11-04

**Soundness:** 2
**Presentation:** 1
**Contribution:** 2
**Rating:** 2
**Confidence:** 2

**Summary:**

The paper studies dynamic Bayesian optimization. In this problem, there is an unknown function f(x,t) that varies with space (x) and time t. We have a Bayesian prior on this function.  At each time t, an algorithm chooses an x_t based on previous observations x_s, f(x_s, s) + noise for s<t.  The paper does not seem to explicitly state the goal but I believe it is to choose a sequence x_t, t=1,...,T to maximize the expected value of E[ \sum_{t=1}^T f(x_t, t)].

**Strengths:**

In many important problems, the objective function varies over time. It is important to develop good algorithms for such settings.

**Weaknesses:**

#1. The number of benchmark methods is small. Only two benchmark methods were used, GP-UCB and WDBO. This makes it hard to judge whether the proposed method is SOTA.  The paper writes that the other methods from the literature review were not used as benchmarks because open-source code is not available.

Two pieces of related work with code are https://github.com/brunzema/et-bo and https://github.com/brunzema/uitvbo.  The first repository reproduces the experiments in https://arxiv.org/abs/2208.10790, which include R-GP-UCB (which is the submitted paper's lit review but not considered in their experiments), ET-GP-UCB (the method in this uncited paper), and two other benchmarks: TV-GP-UCB and UI-TVBO.  To be competitive at ICLR, more work and attention should be put in to evaluation.


#2. The paper is not clearly written.  For example:
- The way in which performance is measured is not clearly stated when the problem is defined. I am only inferring the goal based on the fact that mean regret is reported as a the performance measure in the appendix.
- What is o-tilde?  I understand it is the "desired" outcome but what does this mean?  And what is its relationship to pi?
- What is boldface C on line 171?  This isn't defined.
- The notation switches from BO-style notation in section 2.1 to POMDP-style notation in section 2.2. It would be better to use one notation style. This should be feasible in a 9-page paper.
- In Assumption 1, when it writes S_t = disc(GP_t), does the paper really mean "="?  Perhaps it mean that S_t is an element of disc(GP_t).  This Assumption could be stated much more clearly without defining two new pieces of notation (GP_t and disc).  And what does GP_t mean down in Assumption 2?  Is it still a vector?  It is stated that it is an element of N(0,1).  Vectors are not typically thought of as being elements of probability distributions.  Perhaps this is also a typo?
- n is used to indicate both a number of observations and the "number of discrete inputs for each dimension of the GP excluding the time" (line 212).
- The paper writes a POMDP description of a BayesOpt problem. It writes on lines 210-11 that the state is S_t = ((x_1, t), ..., (x_n, t)) and the observation is O_t = (y_1, ..., y_n), where I believe that n is the number of observations that have been made by time t. Typically, you would include the past observations into the state. As written, in equations (11) and (12), where it references the conditional distribution of S_{t+1} | O_t, one must reason in a complex way about the what the historical x_i might have been given the policy and the observed y_i.
- In the above notation for S_t, it is overkill to include many copies of t in S_t.  Just one is enough.

#3. The paper discretizes the search space (Assumption 1) in a way that presumably loses a great deal of information that would otherwise be present if the state were handled continuously.

**Questions:**

It would clarify my understanding of the paper if you could address my questions above under #2.

---

### Meta-Review · Area_Chair_pCsC · 2026-01-06

**Summary:**

This paper proposes a new acquisition function for dynamic / time-varying Bayesian optimization derived from free energy principles, with the stated goal of improving tracking of moving optima under query restrictions. All reviewers find the topic and high-level idea interesting, but there is broad agreement that the current version does not yet meet ICLR standards due to clarity issues and insufficient empirical support.

**Reviewer Concerns:**

No rebuttal was provided.

Outstanding concerns:

* Empirical evaluation is limited (primarily against GP-UCB and WDBO), and reviewers explicitly point out that other relevant TVBO baselines appear feasible to include (with available implementations) but are missing. This makes it hard to assess the method’s competitiveness.

* BOBA’s key parameter is tuned per-problem, while the corresponding choices/tuning for UCB are unclear or not reported.

* There appear to be severe presentation issues (e.g., undefined notations; inconsistent notation between BO and POMDP context, confusing statements in Assumptions 1/ 2, and general vagueness in key definitions).

* The method appears to require (i) problem-specific tuning in a "zero-shot" BO setting and (ii) information that may be unavailable in true black-box scenarios (e.g., prior knowledge about the optimum/ preferred observations).

* Other issues: Negative regret; ambiguity about whether the GP-UCB baseline is time-aware (access to time coordinate). Some missing citations and code issues.

**Reviewer Scores:**

n/a

---

### Decision · Program_Chairs · 2026-01-26

Reject